**483**

# Conventional rigid 2D substrates cause complex contractile signals in monolayers of human induced pluripotent stem cell-derived cardiomyocytes

Eline Huethorst[1,2] , Peter Mortensen[3], Radostin D. Simitev[3] , Hao Gao[3], Lotta Pohjolainen[4] ,
Virpi Talman[4], Heikki Ruskoaho[4], Francis L. Burton[1], Nikolaj Gadegaard[2] and Godfrey L. Smith[1]

[1] *Institute of Cardiovascular and Medical Sciences, University of Glasgow, Glasgow, UK*
[2] *Division of Biomedical Engineering, James Watt School of Engineering, University of Glasgow, Glasgow, UK*
[3] *School of Mathematics and Statistics, University of Glasgow, Glasgow, UK*
[4] *Drug Research Program and Division of Pharmacology and Pharmacotherapy, Faculty of Pharmacy, University of Helsinki, Helsinki, Finland*

Edited by: Bjorn Knollmann & Michael Shattock

The peer review history is available in the supporting information section of this article (https://doi.org/10.1113/JP282228#support-information-section).

[Correction made on 20 January 2022, after first online publication: The article has been updated to correct an error in the sequence of the figures and figure legends.]

**Eline Huethorst** obtained her BSc in Life Science and Technology from the University of Groningen and her MSc in Regenerative Medicine and Technology from the University of Utrecht. After completion of her master's degree, she was awarded a 4-year PhD studentship from the British Heart Foundation and moved to Glasgow to do her PhD in cardiac regeneration in the labs of Prof. Godfrey Smith and Prof. Nikolaj Gadegaard. Her current studies focus on the development of small cardiac tissues to investigate the acute post-implantation phase in an *ex vivo* rabbit model to understand the factors that influence integration and electro-mechanical coupling.

The Journal of Physiology

**Abstract** Human induced pluripotent stem cell-derived cardiomyocytes (hiPSC-CM) in monolayers interact mechanically via cell–cell and cell–substrate adhesion. Spatiotemporal features of contraction were analysed in hiPSC-CM monolayers (1) attached to glass or plastic (Young's modulus (E) >1 GPa), (2) detached (substrate-free) and (3) attached to a flexible collagen hydrogel (E = 22 kPa). The effects of isoprenaline on contraction were compared between rigid and flexible substrates. To clarify the underlying mechanisms, further gene expression and computational studies were performed. HiPSC-CM monolayers exhibited multiphasic contractile profiles on rigid surfaces in contrast to hydrogels, substrate-free cultures or single cells where only simple twitch-like time-courses were observed. Isoprenaline did not change the contraction profile on either surface, but its lusitropic and chronotropic effects were greater in hydrogel compared with glass. There was no significant difference between stiff and flexible substrates in regard to expression of the stress-activated genes *NPPA* and *NPPB*. A computational model of cell clusters demonstrated similar complex contractile interactions on stiff substrates as a consequence of cell-to-cell functional heterogeneity. Rigid biomaterial surfaces give rise to unphysiological, multiphasic contractions in hiPSC-CM monolayers. Flexible substrates are necessary for normal twitch-like contractility kinetics and interpretation of inotropic interventions.

(Received 10 August 2021; accepted after revision 3 November 2021; first published online 11 November 2021)

**Corresponding author** Godfrey L. Smith: Institute of Cardiovascular and Medical Sciences, Sir James Black Building, University of Glasgow, Glasgow G12 8QQ, UK. Email: Godfrey.Smith@glasgow.ac.uk

**Abstract figure legend** Spatio-temporal contractility analysis of human induced pluripotent stem cell-derived cardiomyocyte (hiPSC-CM) monolayers seeded on conventional rigid surfaces (glass or plastic) showed variable multiphasic contraction events across the monolayer despite identical action potentials. These multiphasic patterns are not present in single cells, in detached monolayers or in monolayers seeded on soft substrates such as a hydrogel, where only physiological 'twitch'-like transients are observed. A computational model of cell clusters supports the biological findings that partial adhesion of cell clusters to a rigid surface causes multiphasic contractile behaviour. Experimental data showed that hiPSC-CM monolayers on rigid substrates have significantly increased contractile duration and a decreased lusotropic drug response when compared to responses on a flexible substrate. Despite the unphysiological nature of the contractile events on rigid surfaces, there is no indication that the multiphasic patterns are associated with significantly higher activation of the stress-activated signalling pathways.

## Key points

- Spatiotemporal contractility analysis of human induced pluripotent stem cell-derived cardiomyocyte (hiPSC-CM) monolayers seeded on conventional, rigid surfaces (glass or plastic) revealed the presence of multiphasic contraction patterns across the monolayer with a high variability, despite action potentials recorded in the same areas being identical.
- These multiphasic patterns are not present in single cells, in detached monolayers or in monolayers seeded on soft substrates such as a hydrogel, where only 'twitch'-like transients are observed.
- HiPSC-CM monolayers that display a high percentage of regions with multiphasic contraction have significantly increased contractile duration and a decreased lusotropic drug response.
- There is no indication that the multiphasic contraction patterns are associated with significant activation of the stress-activated *NPPA* or *NPPB* signalling pathways.
- A computational model of cell clusters supports the biological findings that the rigid surface and the differential cell–substrate adhesion underly multiphasic contractile behaviour of hiPSC-CMs.

## Introduction

The cyclic pump function of the heart is driven by an electrical signal that originates from the sino-atrial (SA-) node and runs through the cardiac conductive system to reach the ventricular myocardium (Dobrzynski *et al.* 2013). Activation of voltage-sensitive channels on the membranes of cardiomyocytes (CMs), including voltage-sensitive L-type calcium channels, leads to excitation–contraction coupling, entailing the calcium-dependent activation of sarcomeres and cellular contraction (Bers, 2002). Cell contraction force is generated via cell–cell adhesions and cellular anchor-points to the extracellular matrix (ECM) proteins,

including collagen type-1 (Col-1) and fibronectin (FN) (Pandey *et al.* 2018). One of the major determinants of the passive stiffness of the myocardium is the ECM and it contributes to the resistive force during the active shortening phase of the cardiac cycle (Granzier & Irving, 1995). Another component is titin, which anchors the myosin filaments to the Z-disc within cardiac sarcomeres and accounts for 70% of the ventricular passive response (LeWinter & Granzier, 2010).

Myocardial stiffness changes during developmental stages and in disease. Human neonatal myocardium has a Young's modulus of around 10 kPa and this increases over time to approximately 50 kPa for a healthy adult heart (Travers *et al.* 2016; Ramadan *et al.* 2017; Ward & Iskratsch, 2020). Cardiac fibrosis, present in some cardiac disease states, increases this stiffness even further to approximately 100 kPa (Travers *et al.* 2016; Ramadan *et al.* 2017; Ward & Iskratsch, 2020). Stiffening of the myocardium is mainly caused by deposition of more or different types of ECM protein by myofibroblasts, and the resulting alterations in myocardial stiffness affect the functionality/contractility of cardiomyocytes *in vivo* (Li *et al.* 2014). Like other cell types, cardiomyocytes bind to their surrounding ECM and to neighbouring cells through cell adhesion complexes in the cell membrane (Samarel, 2005; Geiger *et al.* 2009; Pandey *et al.* 2018; Santoro *et al.* 2019; Ward & Iskratsch, 2020). Both cell–cell and cell–ECM adhesions sense mechanical changes in the cellular environment and lead to functional changes within the cell via activation of overlapping intracellular pathways (del Rio *et al.* 2009; Yonemura *et al.* 2010; Iskratsch *et al.* 2014; Wickline *et al.* 2016; Pandey *et al.* 2018; Izu *et al.* 2019; Monemian Esfahani *et al.* 2019; Saucerman *et al.* 2019). Studies investigating the effects of substrate stiffness on the contractility of cardiomyocytes have found changes in sarcomere organization (Jacot *et al.* 2008; Rodriguez *et al.* 2011; Heras-Bautista *et al.* 2014; Ribeiro *et al.* 2015, 2020), myofibril formation and/or organization (Engler *et al.* 2008; Feaster *et al.* 2015; Ribeiro *et al.* 2015), calcium handling (Jacot *et al.* 2008; Rodriguez *et al.* 2011; Boothe *et al.* 2016; van Deel *et al.* 2017) and force generation (Rodriguez *et al.* 2011; Ribeiro *et al.* 2015, 2020). However, these studies were mostly done with single cells, and therefore omitted the additional role of cell–cell coupling and intercellular force transmission, which could be important for contractile dynamics across the monolayer (Yonemura *et al.* 2010; Monemian Esfahani *et al.* 2019).

Human induced pluripotent stem cell-derived cardiomyocytes (hiPSC-CMs) are used in laboratories for basic and commercial research including cardiotoxicity studies and regenerative medicine (Blinova *et al.* 2018; van Meer *et al.* 2019). HiPSC-CMs are usually grown in a monolayer format on standard tissue culture plastic (TCP) or glass substrates, which have a Young's modulus greater than 1 GPa (Travers *et al.* 2016), 3–4 orders of magnitude stiffer than the native myocardium. To ensure cell adherence to the underlying structure, the plastic or glass surface is coated with a thin layer of ECM components such as fibronectin (Blinova *et al.* 2018). These coatings do not alter the overall stiffness of the cell–surface interface, which thus remains inappropriately high for the normal function of the hiPSC-CMs and can result in the loss of hiPSC-CM contractility (Heras-Bautista *et al.* 2014). Nonetheless, 2D cultures on glass or plastic substrates are still the preferred platform for toxicology assays, because they are the most practical and cost-effective (Guth *et al.* 2019; Ribeiro *et al.* 2019).

While other studies have looked at the relation between substrate stiffness and cardiac differentiation, contraction force or calcium handling, no other study has looked into the effect of substrate rigidity on the contraction time-course of hiPSC-CM monolayers. The present study investigates the contraction time-courses of hiPSC-CM monolayers on routinely used stiff plastic or glass substrates and compared these with the time-courses of hiPSC-CMs seeded on substrates with stiffness characteristics more typical of the native myocardium. These experiments were done using a novel method to analyse contractility in a spatiotemporal manner and were paralleled by a bioassay for cardiac mechanical stress. *In silico* modelling was used to test the mechanistic hypothesis generated to explain the effect of substrate stiffness on muscle contraction.

## Materials and methods

### Preparation of silicone stencils

Stencils were used to create monolayers with a diameter of 3 mm. Stencils were made using silicone (Sylgard-184) and bespoke moulds. The lumen diameter was 1, 2 or 3 mm, and the outer diameter was either 4 mm (thin stencils) or 6 mm (thick stencils). Prior to cell culture, stencils were sterilized using 70% EtOH followed by ultraviolet light for 20 min.

### HiPSC-CM culture

Two different commercially available hiPSC-CMs were used: ICell[2] (Cellular Dynamics International (CDI), Cat# R1218, purity >95% cardiomyocytes) and Cor.4U (NCardia, purity 89% cardiomyocytes (Huo *et al.* 2017)). Cells were seeded on glass, TCP or hydrogel substrates that were coated with FN (10 µg/ml, bovine, Gibco, Cat# 33010018) in either an entire well of a 96-well plate or inside a stencil placed in the well. The seeding density advised by the manufacturers differs between ICell[2] and Cor.4U hiPSC-CMs, but this also depended on the experimental needs. Exact cell densities can be

**Table 1. An overview of the number of plated and viable cells for various cell-seeding densities and stencil sizes**

| Plated cells | | Cell type | | | | | |
| --- | --- | --- | --- | --- | --- | --- | --- |
| | | Cor.4U (NCardia) | | | ICell$^2$ (CDI) | | |
| Stencil size | Area (mm$^2$) | Number of plated cells per well | | | Number of plated cells per well | | |
| | | 1× | 2× | 4× | 1× | 2× | 4× |
| Full well | 31.2 | 30,000 | 60,000 | 120,000 | 50,000 | 100,000 | 200,000 |
| 3 mm | 7.1 | 6803 | 13,605 | 27,211 | 11,338 | 22,675 | 45,350 |
| 2 mm | 3.1 | 3023 | 6047 | 12,094 | 5039 | 10,078 | 20,156 |
| 1 mm | 0.8 | 756 | 1512 | 3023 | 1260 | 2520 | 5040 |

| Viable cells | | Cell type | | | | | |
| --- | --- | --- | --- | --- | --- | --- | --- |
| | | Cor.4U (NCardia) | | | ICell$^2$ (CDI) | | |
| Stencil size | Area (mm$^2$) | Number of viable cells per well | | | Number of viable cells per well | | |
| | | 1× | 2× | 4× | 1× | 2× | 4× |
| Full well | 31.2 | 15,900 *(53%)* | NA | NA | 26,500 *(53%)* | NA | NA |
| 3 mm | 7.1 | 3878 *(57%)* | 5306 *(39%)* | 7347 *(27%)* | 6463 *(57%)* | 8843 *(39%)* | 12,245 *(27%)* |
| 2 mm | 3.1 | 1451 *(48%)* | 5986 *(44%)* | 3024 *(25%)* | 2419 *(48%)* | 4434 *(44%)* | 5039 *(25%)* |

The cell densities used depended on the stencil size and the commercially available hiPSC-CMs cell line. Cell-seeding densities were calculated according to the manufacturers protocol and with the normal cell density '1×' being 100,000 cells/cm$^2$ for Cor.4U (NCardia) and 150,000 cells/cm$^2$ for ICell$^2$ (CDI). The number of plated cells was based on the stencil size and cell-seeding density. The number of viable cells was based on the percentages of viable cells as presented in Figure 3E. This study was only done with Cor.4U hiPSC-CMs (NCardia). For ICell$^2$ hiPSC-CMs (FCDI) it was assumed that the percentage of viable cells would be similar, and thus the same percentages were used to calculate the number of viable cells.

found in Table 1; in this paper, '1×' cell density denotes the normal surface cell density per cm$^2$ as advised by the manufacturer of the hiPSC-CMs, namely $1.5 \times 10^5$ cells/cm$^2$ for ICell$^2$ and $1 \times 10^5$ cells/cm$^2$ for Cor.4U hiPSC-CMs. Twice and four times the normal surface cell density are denoted by '2×'and '4×', respectively. An overview of cell seeding using stencils is shown in Fig. 3A. HiPSC-CMs were maintained in a controlled environment with 5% $CO_2$ and at 37°C. Stencils were removed 2 days after plating, and the medium was changed every 2–3 days. HiPSC-CM maintenance medium was provided by the hiPSC-CM manufacturer.

## Voltage recordings

HiPSC-CMs were incubated in serum-free medium (BMCC: $CaCl_2$ 1.49 mM, $MgSO_4*7H_2O$ 0.81 mM, KCl 4.4 mM, $NaHCO_3$ 36 mM, NaCl 77.59 mM, $Na_2HO_4P$ 0.91 mM, $Na_2SeO_3$-$5H_2O$ 0.0001 mM, $KNO_3$ 0.0008 mM, D-glucose 25 mM, sodium pyruvate 1 mM) for at least 1 h and subsequently loaded with FluoVolt Dye (1:1000, Invitrogen, Cat# F10488) and Powerload Concentrate (1:100, Invitrogen, Cat# F10488) for 20 min at 37°C. Action potentials (APs) were recorded on the CellOPTIQ system (Hortigon-Vinagre *et al.* 2016) using a 40×

objective, 470 nm LED, photomultiplier tube (PMT) and a sensor with an acquisition rate of 10 kHz. Plates were placed in an on-stage incubator during the experiment, maintaining 5% $CO_2$ and 37°C. APs were analysed using CellOPTIQ software, which computed the time-averaged APs ($n > 3$) and calculated the beating frequency, depolarization time ($T_{Rise}$) and AP duration at 90% of the amplitude ($APD_{90}$) (Fig. 1A).

## Intracellular Ca$^{2+}$ recordings

HiPSC-CMs (ICell$^2$, CDI) were incubated in serum-free medium (BMCC) for at least 1 h. Subsequently, cells were loaded with 1 $\mu$M Cal520-AM (AAT Bioquest, Cat# 21130) and 0.02% Pluronic acid-F127 (Biotium, Cat# 59000) for 20 min at 37°C. Calcium transients were recorded on the CellOPTIQ system (Hortigon-Vinagre *et al.* 2016), similar to that of voltage recordings as described in the paragraph above. In brief, a 40× objective, 470 LED, PMT was used, and cells were kept in a controlled environment with 5% $CO_2$ and 37°C during recordings. Calcium transients were analysed using the CellOPTIQ software, where the spontaneous beating frequency, $T_{Rise}$, $CaT_{50}$ and $CaT_{90}$ were calculated as explained in Fig. 1B.

## Contractility recordings

Bright-field video recordings were made using a high-speed camera (Hamamatsu ORCA-flash 4.0 V2 digital CMOS camera C11440-22CU) (100 fps, 600 × 600 pixels) and a 4×, 10× or 40× objective (Olympus, air objectives). Video frames were analysed using the open-source MUSCLEMOTION (MM) contractility algorithm published by Sala *et al.* (2018), which measures movement as a function of pixel intensity and has been verified against a number of other measures of mechanical function. Using the MM algorithm, contractility traces were time-averaged ($n > 3$) and analysed for the beating frequency, contractile amplitude, contraction time ($T_{Contraction}$) and relaxation time ($T_{Relaxation}$), and the contractility duration at 50% of the peak ($CD_{50}$) was measured (Fig. 1*C*). MM can be used in a range of illumination conditions without affecting the temporal parameters. However, the contractile amplitude is correlated with the pixel intensity/illumination levels, which makes comparisons between different groups/substrates difficult to interpret and they are therefore not included.

Additionally, video frames were subdivided into grid squares, and the MM algorithm was applied on each grid square, to spatially compare hiPSC-CM contractility variations. Additional measurements were derived from the spatiotemporal analysis: the number of peaks within one contraction and the start time ($T_{Start}$), measured as the time to 50% of the amplitude during the contraction phase. Dispersion (spread) of $T_{Start}$ and $CD_{50}$ values was estimated as the difference between 10th and 90th percentiles ($IP_{90}$), to exclude outliers (Fig. 7*H*).

## Assessing morphology and functionality of hiPSC-CM monolayers

On day 2 after stencil removal (day 4 after cell seeding), hiPSC-CMs were assessed for contractility and voltage as described above. Subsequently, cells were fixed with 4% paraformaldehyde and stained for F-actin (Rhodamine Phalloidin, 1:50, Invitrogen, Cat# R415) and DNA (DAPI, 1:1000, ThermoFisher Scientific, Cat# 62247). The full area of a 96-well plate was imaged using EVOS-2FL auto with a 10× objective, and individual images stitched together automatically by the EVOS software. Stitched images were analysed using CellProfiler software (v3.0.0).

## Cell-sheet detachment studies

HiPSC-CMs were cultured for 4 days prior to the start of the experiment. On day -4, 3 mm stencils were placed onto fibronectin-coated thermosensitive PIPAAm dishes (Nunc Dishes with UpCell Surface, Thermo-Fisher Scientific, Cat# 174903) and Cor.4U hiPSC-CMs (NCardia) were seeded inside the stencils in a 2× cell density ($2 \times 10^5$ cells/cm$^2$). The stencil was removed after 2 days (day -2) and cultured for two more days. On day 0, cells were cooled to 25°C for 1 h under culture conditions with 5% $CO_2$ and allowed to detach from the surface of the dish. Then, cell sheets were lifted off the dish using a pipette. If the sheet was stuck to the plate, it was carefully detached by pipetting medium underneath the cell sheet. After detaching, hiPSC-CM sheets were transferred to a new fibronectin-coated standard TCP dish and incubated without medium for 10 min, followed by addition of maintenance medium. Recordings for contractility were made on day 0 at 37°C, at 25°C and 2–3 h after detaching, and on days 1, 3, 5 and 7. Here, a 10× objective was used to maximize the field of view. Grid-square analysis was done with 30 × 30 grid squares, each of approximately 40 × 40 $\mu$m. Additionally, voltage recordings (40× objective) were made on day 7.

## Preparation of recombinant collagen-like peptide hydrogel

Photo cross-linkable recombinant human collagen-like peptide (RCP), was kindly provided by FujiFilm Manufacturing Europe B.V. (Tilburg, The Netherlands) and was used to make hydrogels with a Young's modulus

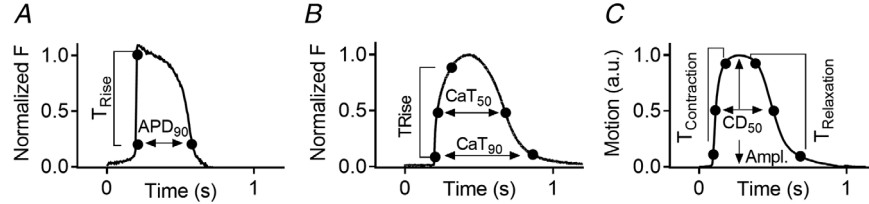

**Figure 1. Explanation of physiology parameters for (*A*) electrophysiology, (*B*) calcium, and (*C*) contraction**

$T_{Rise}$ = time from 10 to 90% of amplitude. $APD_{90}$ = action potential duration at 90% of the amplitude. $CaT_{50}$ = calcium transient duration at 50% of the amplitude. $CaT_{90}$ = calcium transient duration at 90% of the amplitude. $T_{Contraction}$ = contraction time. $T_{Relaxation}$ = relaxation time. Ampl. = amplitude. $CD_{50}$ = contraction duration at 50% of the amplitude.

(E) of approximately 22 kPa (unpublished observation, FujiFilm Manufacturing Europe B.V.) (Tytgat *et al.* 2019). First, RCP was dissolved in PBS$^{+Mg/+Ca}$ (10% w/v) and incubated at 37°C for 1 h. Lithium-phenyl (2,4,6-trimethylbenzoyl) phosphinate (LAP, TOCRIS, Cat# 6146) was used as a cross-linker and dissolved in PBS$^{-/-}$ (2.5% w/w) and subsequently incubated at 37°C and protected from light for at least 20 min until fully dissolved. RCP, LAP and FN were mixed thoroughly in a ratio of 1000:26:50, respectively, and added to a mould (6 mm Ø × ~300 $\mu$m thick), followed by cross-linking using a 365 nm LED at 10 mW/cm$^2$ for 5 min in the absence of oxygen. After cross-linking, hydrogels were incubated overnight in 100 $\mu$g/ml FN (bovine, Gibco, Cat# 33010018) in PBS$^{+Mg/+Ca}$ at 4°C. Prior to cell-seeding, gels were soaked in culture medium for 1 h.

## RCP hydrogel experiments – cell-seeding and functionality assessment

On day -2, hydrogels were placed on glass dishes (MatTek, Cat# P35G-1.5-14-C), and 3 mm stencils were placed on top of the hydrogels. Fibronectin-coated glass substrates were used as a control. ICell$^2$ hiPSC-CMs (CDI) were seeded in 2× cell density inside the stencils. Stencils were removed on day 0 and maintenance medium was added. Contractility recordings were made on days 0 (2–3 h after stencil removal), 1, 3, 5 and 7 using a 4× objective. Additionally, voltage recordings were made on day 7.

## *β*-adrenergic stimulation of hiPSC-CMs

On day -2, hiPSC-CMs (CDI) were seeded in the 2× cell density on hydrogels (6 mm Ø × ~300 $\mu$m thick) inside the 3 mm stencils and 2 days later (day 0) the stencils were removed. On day 5, cells were stimulated with isoprenaline (ISO, Sigma-Aldrich, Cat# I5627), a *β*-adrenoreceptor agonist that has a positive inotropic effect in cardiac cells. First, baseline contractility was recorded, after which either ISO (300 nM) or a vehicle (culture medium) was added. After 5 min of incubation, contractility recordings at either a spontaneous rate or a fixed rate of 1 Hz were made. Here, hiPSC-CMs seeded on glass substrates (MatTek) served as a control substrate.

## RNA isolation and qRT-PCR

Expression of natriuretic peptide A (NPPA) and natriuretic peptide B (NPPB) mRNA was studied as described previously (Pohjolainen *et al.* 2020). Endothelin-1 (ET-1) upregulates the expression of *NPPA* and *NPPB* in hiPSC-CMs and was therefore used as a positive control (Pohjolainen *et al.* 2020). Cells were lysed

in 400 $\mu$L of Trizol reagent (Invitrogen, Carlsbad, CA, USA) and stored at -80°C. Total RNA was isolated using the Phase Lock Gel Heavy tubes (Quantabio, Beverly, MA, USA) according to the manufacturer's instructions and RNA concentration and quality were analysed using a NanoDrop 1000 (Thermo Fisher Scientific, Waltham, MA, USA) spectrophotometer. cDNA was synthesized from 50–250 ng of total RNA in 10 $\mu$L reactions with a Transcriptor First Strand cDNA Synthesis Kit (Roche, Basel, Switzerland) according to the manufacturer's protocol using random hexamer primers and an MJ Mini Personal thermal cycler (Bio-Rad). The cDNA was diluted 1:9 in PCR-grade H$_2$O, of which 4.5 $\mu$L was used for qPCR analysis. Commercial TaqMan Gene Expression Assays for *NPPA* (Hs00383230_g1), *NPPB* (Hs01057466_g1), *ACTB* (4333762F) and eukaryotic 18S rRNA (4352930E), all from Thermo Fisher Scientific, were used with LightCycler 480 Probes Master reagent (Roche) according to the manufacturer's instructions. Analysis was performed on a LightCycler 480 Real-Time PCR System (Roche). Two technical replicates were used for each reaction, and the average of technical replicates was used in the analysis. To confirm absence of PCR contamination, no-template controls were used. The ΔΔCt method was used to analyse the relative gene expression. First, the quantification cycle (Cq) values of *NPPA* and *NPPB* were normalized to the average of the Cq values of reference genes *ACTB* and 18S rRNA of the same sample, after which the obtained ΔCq values were normalized to the ΔCq values of the control sample.

## Mathematical model

A mathematical model was formulated to help interpret and explain the spatiotemporal features of contraction measured in the experiments. As a mathematical model of our experiments with contraction on a substrate, we consider a 1D chain of contractile units connected in serial. The units represent cells and sit on a horizontal frictionless surface representing the substrate. The two ends of the chain are connected to the surface by linear elastic springs. Illustrations of this setup are shown in Fig. 2*C* and Fig. 13*A*. To vary the stiffness of the substrate in this model we consider different values of the linear string constants.

The following scheme proposed by Timmermann *et al.* (2019) is used to connect the contractile units and the end springs into a chain:

$$\frac{dL_i}{dt} = \left( 2\frac{d\widehat{L}_i}{dt} - \frac{d\hat{L}_{i-1}}{dt} - \frac{d\hat{L}_{i+1}}{dt} \right) / 2,$$
$$\forall i \in \{2, 3, \ldots, N-1\},$$

$$\frac{dL_1}{dt} = \left(\frac{d\hat{L}_1}{dt} - \frac{d\hat{L}_2}{dt}\right)/2,$$

$$\frac{dL_N}{dt} = \left(\frac{d\hat{L}_N}{dt} - \frac{d\hat{L}_{N-1}}{dt}\right)/2,$$

where $L_i$ is the length of the contractile unit $i$ at a given timestep, with $L_1$ and $L_N$ being the elastic springs at the left and right end, and $N$ is the total number of units (including the springs). The hats ^ denote quantities at the preceding timestep and are computed using the myofilament contraction model of Rice *et al.* (2008),

$$\frac{d\hat{L}_i}{dt} = \frac{p + \left(L_i - L_{i,0}\right) \times viscosity}{mass}, \text{ where } p = \int_0^t F dt$$

where impulse $p$ is the integral of the total force $F$ acting on a unit over a time interval. Forces are scaled with the peak twitch force of a rat sarcomere of 101.8 kPa (Daniels *et al.* 1984).

For the contracting units $i = 2. N\text{-}1$, the total force is a sum of active and passive forces, $F = F_a + F_p$, as illustrated in Fig. 2C. The passive force has contributions from the titin in the sarcomere and collagen in the extra cellular matrix, see the paper of Rice *et al.* (2008). A linear elastic force has been added to those considered in Rice *et al.* to ensure that each cell can return to its original length after active contraction and represents passive forces in the whole cell unaccounted for by Rice *et al.*; for example, iso-volumetric forces (Fig. 2B). This modification is justified

by the fact that the model of Rice *et al.* is a myofilament model that we adopt as a model of the whole cell and is critical for reproducing the recorded motion of the myocyte contraction. The active force is taken to be identical to that specified in the Rice model (Rice *et al.* 2008) and is dependent on the calcium concentration. As input calcium profiles we use curves fitted to the recorded calcium concentration in the experimental work, as seen in Fig. 2A. The maximum and minimum concentrations used are taken from the Shannon *et al.* (2004) model of a rabbit ventricular myocyte, as also used in the Rice model.

For the linear elastic springs $i = 1$ and $i = N$, the total force is given by Hooke's law, $F = F_S$, with

$$F_S = k\left(L_i - L_{i,0}\right),$$

where $L_i$ is the current length of the spring, and $L_{i,0}$ is the initial length of the spring. The value $k$ is the stiffness of the spring, or the spring constant. This value is a unit-normalized force per $\mu m^{-1}$. As mentioned, the forces in the Rice model (Rice *et al.* 2008) are normalized relative to the peak twitch force of a rat sarcomere of 101.8 kPa (Daniels *et al.* 1984), thus $k = 1$ represents a stiffness of 101.8 kPa. Although it is understood that stiffness of the stiff plastics or glass would have a stiffness in the order of GPa, $k = 1$ was found to be sufficient to have little movement in the springs, as desired. A more complete discussion and derivation of the model equations is provided by Mortensen (Mortensen, 2021).

The model was tested for variations in the number of contractile units ($N = 3$, $N = 5$ or $N = 10$)

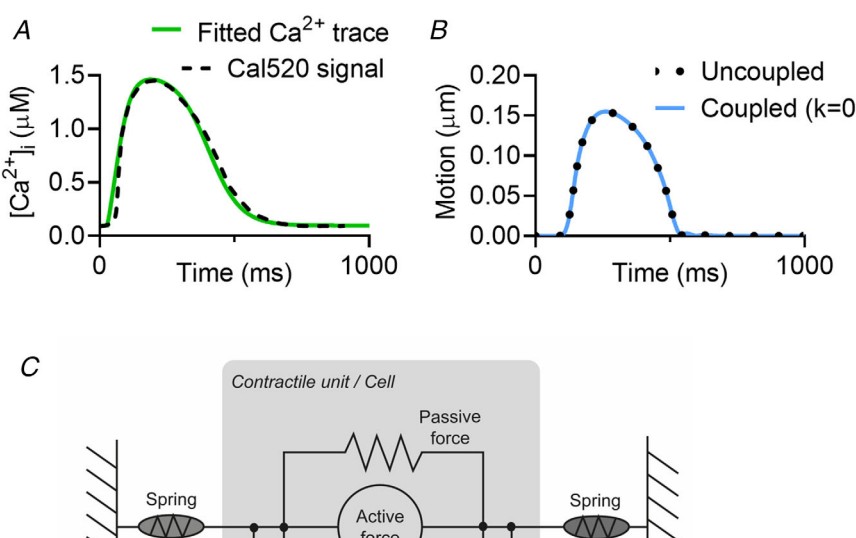

**Figure 2. Mathematical model specifics**
*A*, calcium trace from hiPSC-CMs (Cal520 signal) (continuous line) *vs.* the calcium profile used in the mathematical modelling (broken line). [Correction made on 20 January 2022, after first online publication: 'Cal590' has been corrected to 'Cal520' in the preceding sentence.] *B*, the motion trace from a single contracting unit in the uncoupled system (dotted line) from the model from Rice *et al.* (2008) compared with the motion traces from a single contracting unit, coupled between two springs with $k = 0$, from the model directly from Timmerman *et al.* (2019) (solid blue line). *C*, a schematic of the force components in the contractile units, based on Fig. 1D from Rice *et al* (2008). [Colour figure can be viewed at wileyonlinelibrary.com]

and for the cell-to-cell variation of intracellular calcium ($[Ca^{2+}]_i$) (Cerignoli *et al.* 2012) on the contractile profiles. The variations in $[Ca^{2+}]_i$ tested were 0% (identical), 95–100% (minimal), 75–105% (low), 50–115% (medium) and 25–125% (high), where the maximum calcium level was randomly sampled from a uniform distribution and randomly assigned to each contractile unit.

### Statistical analysis

For each experiment, data was collected from three individual platings (biological replicates) that included multiple wells per group (technical replicates). When using the 40× objective, recordings were made from random locations within each well. The *n*-numbers of each experiment ($n_{Experiment}$) and the total number of samples or wells ($n_{Samples}$) are stated in the figure legend. For statistical analysis, Graphpad Prism v8.0.2 was used and statistical significance was tested using either a nested

*t* test or a nested one-way ANOVA with a Dunnett's *post hoc* test, unless indicated otherwise. The results are presented as means ± standard deviation (SD) together with the raw data points.

## Results

### Small monolayer morphology dependent on stencil size and cell-seeding density

Small monolayers of millimetre-scale (1, 2 and 3 mm) and a variety of cell-seeding densities (1×, 2× and 4× the normal cell density) were assessed prior to the main part of this study, of which example images are shown in Fig. 3*B*. Two and 3 mm stencils seeded with 2× or 4× the cell densities showed similar cell coverage as the control group (Figs. 3*C*–3*E*). From these data it was decided to continue with the 3 mm monolayers.

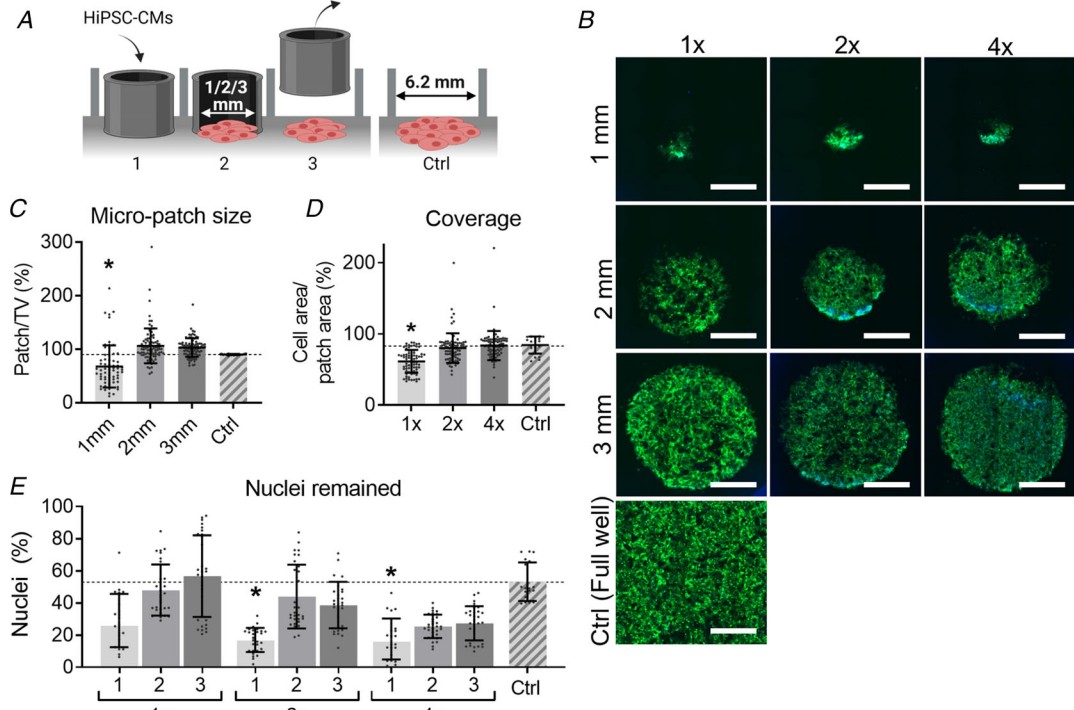

**Figure 3. Assessment of the morphology of small cardiac patches**
*A*, method of creating 2D cardiac sheets using silicone stencils. (1) Place the stencils of various sizes (1, 2 or 3 mm) inside the well and seed hiPSC-CMs in the stencil in desired cell density (normal (1×), double (2×) or quadruple (4×) density). (2) Culture the hiPSC-CM for two days. (3) Remove the stencil, add 200 $\mu$L maintenance medium to every well and culture the patches for two more days. Assessment is done on day 4 after seeding. (4) A well of a 96-well plate (TCP) was seeded with the normal cell density and was used as control (Ctrl). *B*, F-actin (green) and DAPI (blue) staining for hiPSC-CM micro-patches with different sizes and cell densities. Scale bar represents 1 mm. *C*, CellProfiler data showing the relative patch size compared with the theoretical value (TV). *D*, CellProfiler data showing the area coverage by hiPSC-CM calculated by dividing the cell area by the patch area. *E*, CellProfiler data showing the remaining percentage of nuclei, which was calculated as the number of cells counted divided by the number of cells seeded. Results from Cor.4U (NCardia) hiPSC-CM only. Statistical analysis was done using a nested one-way ANOVA using a Dunnet's *post hoc* test and groups were tested against Ctrl. $n_{Plating}$ = 3; $n_{Recordings}$ = 6 to 10 per plate. Absolute values of the *P* values for all comparisons (significant and non-significant) are listed in the online Statistical Supplement. [Colour figure can be viewed at wileyonlinelibrary.com]

## Electrophysiology and contractility parameters do not significantly differ between hiPSC-CM cell lines

Next, the electrophysiology and contractility of both the Cor.4U and ICell[2] hiPSC-CMs were assessed and compared for all three cell-seeding densities. All cell densities showed the same spontaneous beating frequency, electrophysiology and contractility (Figs. 4A and 4B, respectively). As there were no major morphological or physiological differences between the 2× and 4× cell densities, double cell density was used in all subsequent experiments. In addition, differences were observed between the two hiPSC-CM cell lines in spontaneous beating frequency (Fig. 4Aa), APD$_{90}$ (Fig. 4Ac), T$_{Contraction}$ (Fig. 4Ba) and T$_{Relaxation}$ (Fig. 4Bb).

## Significant variation in contractility and negligible variation in voltage across a monolayer

Figure 5 shows typical contractility and voltage signals from hiPSC-CMs seeded as a monolayer on TCP. Fluorescence signals (voltage) and video images (contractility) were recorded in sequence from three discrete regions of approximately $300 \times 300 \mu$m, as shown in Fig. 5A. Segments of these recordings are shown in Fig. 5B. The voltage traces from each location shown in Figs. 5Ba–5Bc illustrate comparable profiles with a very similar shape and duration across the monolayer. In contrast, the contractility traces in the three locations

are different, showing complex time-courses with single or multiple peaks, which return every contraction cycle and are constant beat to beat (Figs. 5Bd–5Bf). This is highlighted in the overlaid plots of Figs. 5C and 5D. These data suggest that the complex contractile profile is not a consequence of non-uniform electrical coupling, but rather is mechanical in origin.

## Spatiotemporal analysis of contractile behaviour of hiPSC-CM reveals variations of movement in adjacent fields of view

The complex contractile behaviour is also visible from video recordings (see Supplementary videos S1 and S2). There are areas in the video frame where the hiPSC-CMs seem to contract twice (centre) next to areas where the cells seem to contract only once (left and right sides). To shed light on this contractile behaviour, the video frame was divided into $3 \times 3$ grid squares, each of $100 \times 100 \mu$m, as shown in diagram Fig. 5E. The MM algorithm was then applied to each grid square, resulting in nine different but simultaneous contractility traces. In Fig. 5F, all nine traces generated by the MM algorithm are displayed together, showing the synchronicity at the start of the contractile phase, followed by contractile profiles that differed in shape. In Fig. 5G, all nine traces are placed in correspondence to their position in the $3 \times 3$ grid. The simple, 'twitch-like' transients occurred adjacent to

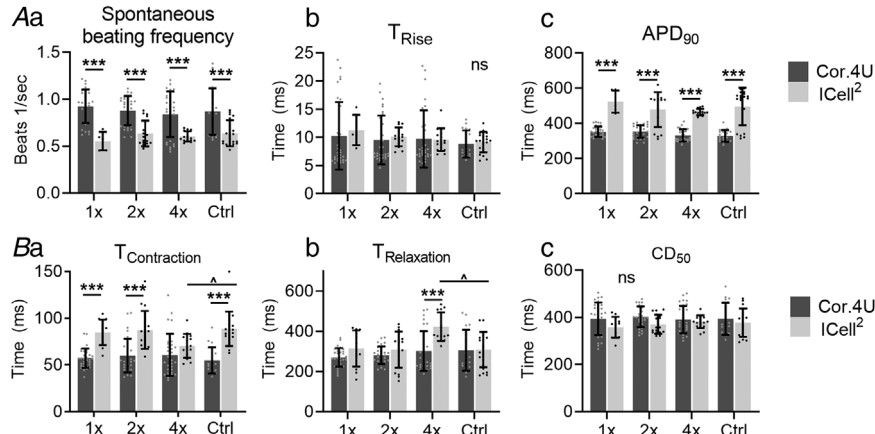

**Figure 4. Differences in electrophysiology and contractility between cell-seeding densities and commercial hiPSC-CM cell lines**
*A*, electrophysiology of hiPSC-CMs showing spontaneous beating frequency (*Aa*) T$_{Rise}$ (*Ab*) and APD$_{90}$ (*Ac*) for both NCardia (black) and ICell[2] (grey) hiPSC-CM cell lines. *B*, contractile behaviour of hiPSC-CMs shown as contraction time (*Ba*), relaxation time (*Bb*) and CD$_{50}$ (*Bc*) for both Cor.4U (dark grey bars) and ICell[2] (light grey bars) hiPSC-CM cell lines. $n = 5$–19 (ICell[2]) and 9–30 (Cor.4U), $N = 3$ experiments for both cell lines. Cell densities were compared with Ctrl for each cell type using mixed effect two-way ANOVA with Dunnett's *post hoc* test (statistical significance indicated with ^). Differences between NCardia and ICell[2] were tested between a two-way ANOVA with Sidak's *post hoc* test (statistical significance indicated with *). Recordings were made on day 2 after stencil removal. Absolute values of the *P* values for all comparisons (significant and non-significant) are listed in the online Statistical Supplement.

complex transients with multiple peaks. This is also visible in Fig. 5*H*, which shows a heatmap for the number of peaks (a). Likewise, heatmaps for the start time ($T_{Start}$) (b), and the contraction duration at 50% of the amplitude ($CD_{50}$) (c) are shown (Fig. 5*H*). These heatmaps also show that areas with relatively high values for $T_{Start}$ or $CD_{50}$ are located adjacent to areas with relatively low values, emphasizing the local variation in contractility. Lastly, the average contractility transient is shown in Fig. 5*I*, which displays a complex profile that is constant from beat to beat, similar to that seen in Fig. 5*B*. These data suggest

that the contractile behaviour of hiPSC-CM monolayers seeded on rigid substrates exhibit complex contractile patterns that vary locally (over a range of $\sim 100\ \mu$m).

## Cell-seeding density does not affect the contractile complexity

Increasing or decreasing the cell-seeding density did not affect the complexity as shown in Fig. 6. Like the standard (2× cell density), the average contractility traces of the 1× and 4× cell-seeding densities showed a

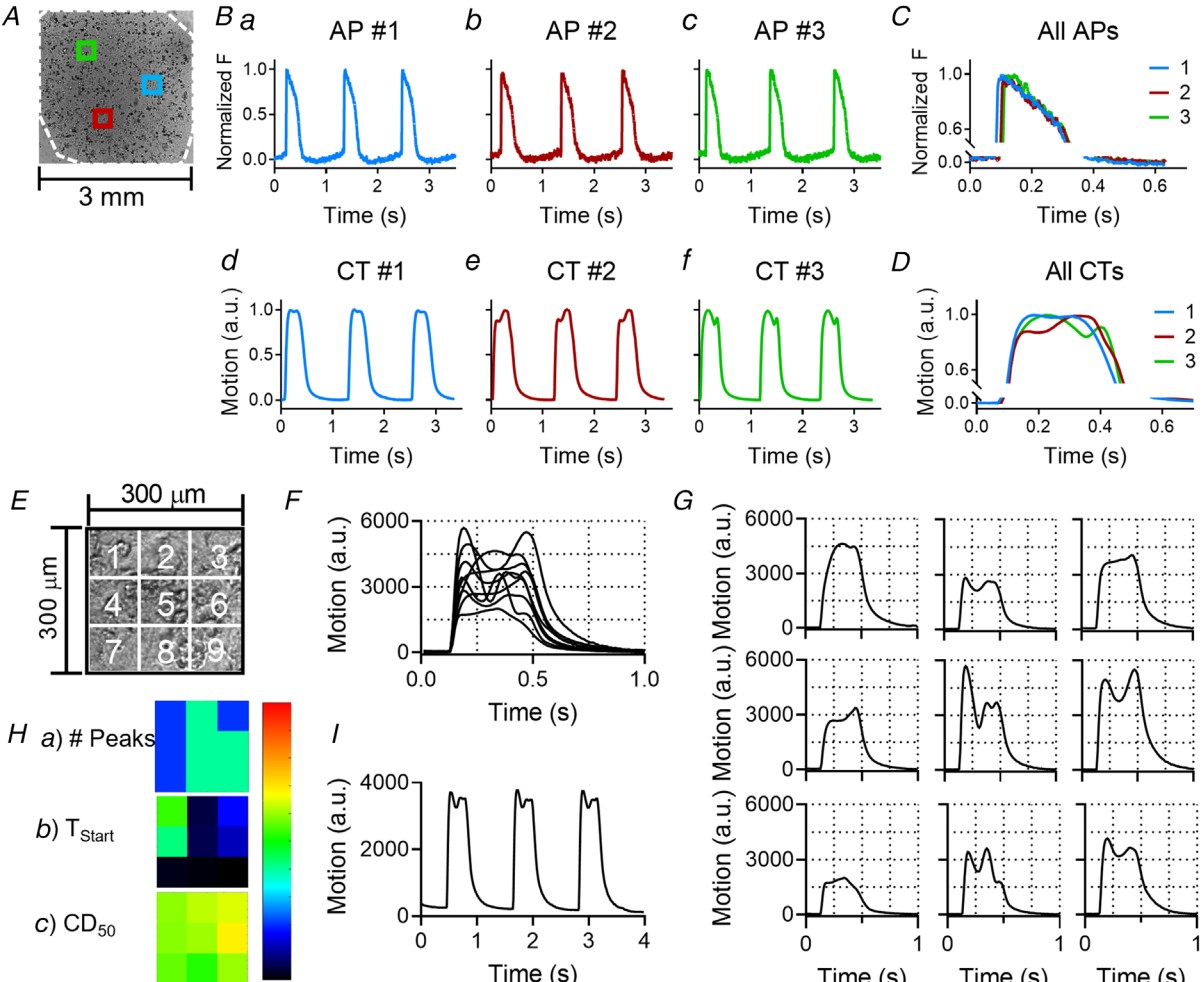

**Figure 5. Significant variation in contractility and negligible variation in voltage across a hiPSC-CM monolayer seeded on tissue culture plastic**
*A*, example of three different locations within the 2D micro-patch indicated with blue, red and green squares. Each location is approximately 300 × 300 $\mu$m. The white dashed line indicates the outer line of the patch. *B*, action potentials (AP, top row, panels a–c) and contractility traces (CT, bottom row, panels d–f) recorded on three different locations within one well (blue, green and red inserts in panel *A*). APs and CTs with the same colour are recorded on the same location. *C*, overlay of the three APs. *D*, overlay of the three CTs. *E*, a 300 × 300 $\mu$m field of view, similar to one of the locations in panel *A*, was subdivided into 3 × 3 grid squares of 100 × 100 $\mu$m each. The MM algorithm was applied to every grid square. *F*, all traces from each grid square (nine in total) were overlapped. *G*, all traces from each grid square were placed in their corresponding location shown in panel *E*. *H*, heatmaps indicating (a) the number of peaks, (b) the start time ($T_{Start}$) and (c) the contractile duration at 50% of the amplitude ($CD_{50}$) (scalebars are 0–5 peaks, 0–40 ms and 0–600 ms, respectively). *I*, the contractility trace taken from the whole area (300 × 300 $\mu$m). [Colour figure can be viewed at wileyonlinelibrary.com]

### A　1x cell density

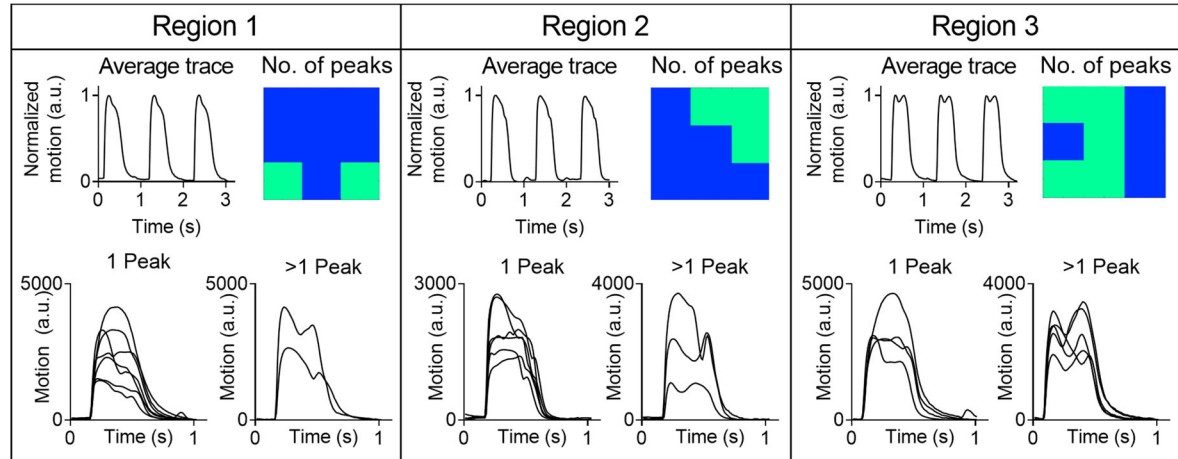

### B　2x cell density (used in the other experiments)

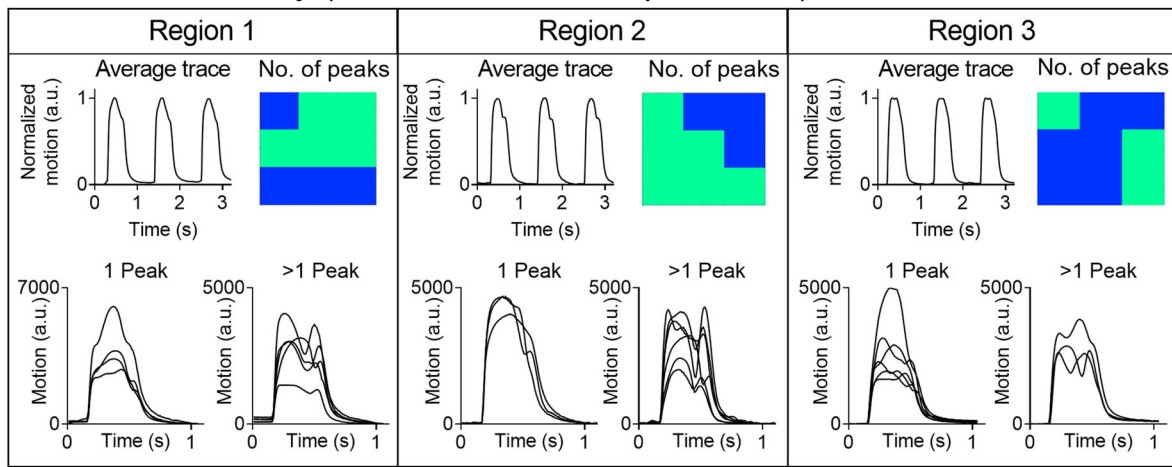

### C　4x cell density

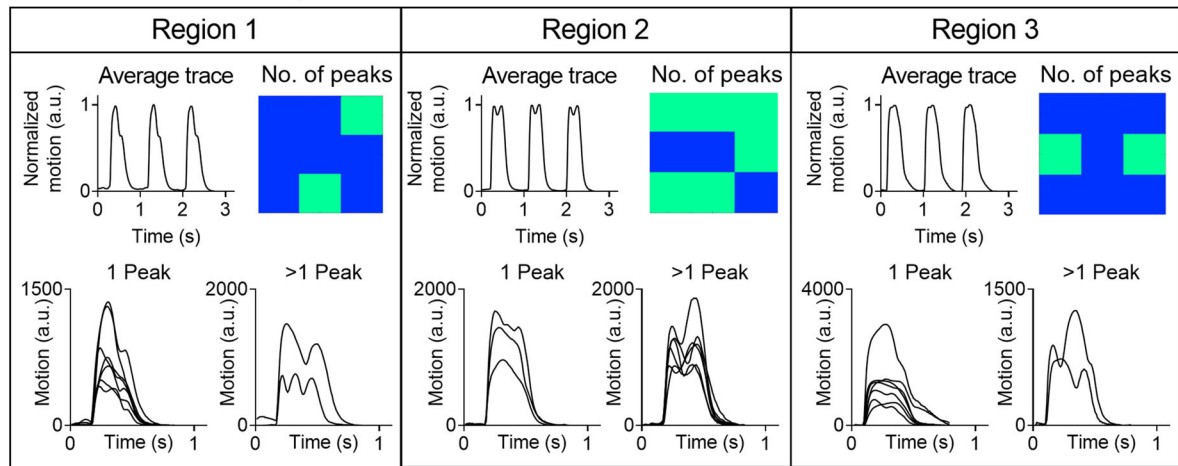

**Figure 6. Cell density does not modulate the complex contractile behaviour on fixed substrates**
HiPSC-CMs seeded in 1× (*A*), 2× (*B*) or 4× (*C*) the cell density suggested by the manufacturer. For each cell density, example traces are shown from three distinct regions of the same micro-patch. These include (1) the average contractile trace recorded from the whole area (top left panel), (2) a heatmap showing the location of grid squares with either 1 (blue) or >1 peak (light green) (top right), (3) the traces with 1 peak (bottom left) and (4) the traces with >1 peak (bottom left). Recordings are from Cor.4U hiPSC-CMs (NCardia) on day 2 after stencil removal. [Colour figure can be viewed at wileyonlinelibrary.com]

complex time-course and were similar to the ones seen in Fig. 5. Spatial analysis revealed similar complex contractile behaviour at all three cell-seeding densities, with both single-peaked and multiple-peaked transients in every region. These data indicate that cell-seeding density does not modulate the complex contractile behaviour on fixed substrates.

### Spatiotemporal contractility analysis across the entire hiPSC-CM monolayer

To further investigate the spatial distribution of the complex contractile behaviour within the entire monolayer of hiPSC-CMs, the entire hiPSC-CM monolayer of 3 mm in diameter was recorded and video frames were subdivided into 30 × 30 grid squares, each 100 × 100 $\mu$m as shown in Fig. 7A. The MM algorithm was then applied to each grid square, resulting in 900 traces which could be categorized as having either 1, 2 or >2 peaks. In Fig. 7B, example traces with either 1, 2 or >2 peaks are shown. In Fig. 7C, all 900 traces of this example are plotted, and Figs. 7D, 7E and 7F show all traces with either 1, 2 or >2 peaks, respectively. Heatmaps like the ones shown in Fig. 7G give an overview of the spatial distribution of the number of peaks, amplitude, $CD_{50}$ and $T_{Start}$. Figure 7H shows an example of the distribution of either $T_{Start}$ or $CD_{50}$ values with the 10th–90th percentile range ($IP_{90}$) annotated, which represents the variability of the measure. This spatial analysis of the complete 3 mm

area of the monolayer was used to characterize the contractile behaviour under a series of conditions.

### The contractile behaviour of hiPSC-CMs before and after detachment from rigid substrates

We tested the hypothesis that the origin of the contractile behaviour is the differential attachment to the underlying substrate by detaching the CMs from the rigid substrate. Figure 8A shows an overview of the experimental procedure. In brief, cells are cultured on the thermosensitive dish, detached from the dish at room temperature, transferred to a new dish on day 0 and allowed to reattach to TCP for a week. The results of this experiment are shown in Figs. 8B–8G.

Figure 8B shows the spontaneous beating frequency throughout the procedure. At day 0, the spontaneous rate was reduced to approximately 50% of control on lowering the temperature to 25°C. On detachment, many samples became quiescent while others returned a normal beating frequency. By day 1 at 37°C, spontaneous activity had returned in the majority of the samples and the rate reduced gradually over the 7 days in culture. The relationship between detachment and the complexity of the contractile event is shown in Fig. 8C–G. Figure 8C indicates that on average 54 ± 15% of the grid squares displayed single-peaked transients under controlled conditions at the start of the experiment. The percentage of single peaks increased towards the

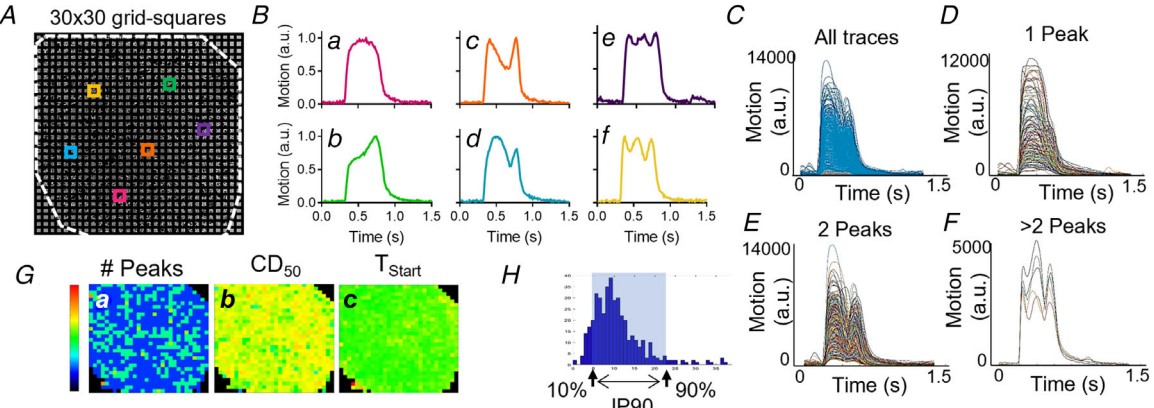

**Figure 7. Spatiotemporal analysis of the entire hiPSC-CM monolayer**
*A*, video frames were subdivided into 30 × 30 grid squares. Each grid square (∼100 × 100 $\mu$m) was analysed using the MM algorithm (33), resulting in 900 individual data points. *B*, various contractile profiles are seen at different locations within one monolayer as represented by the coloured traces (panels *A* and *B*). We observed transients containing one peak (*Ba* and *Bb*), two peaks (*Bc* and *Bd*) and more than two peaks (*Be* and *Bf*). *C*, all traces obtained from one video. *D*, all traces obtained from one video with one peak. *E*, all traces obtained from one video with two peaks. *F*, all traces obtained from one video with more than two peaks. *G*, values for various measurements are plotted in a heatmap. (*a*) The number of peaks (scalebar 1–3 peaks), (*b*) the contraction duration at 50% of the amplitude ($CD_{50}$) (scalebar 0–1200 ms), and (*c*) the start times ($T_{Start}$) (scalebar 0–600 ms). *H*, an example of the distribution of 900 data points following spatiotemporal analysis. From this distribution, the 10th–90th percentile difference ($IP_{90}$) was calculated as illustrated to obtain a more realistic average within one group and thus allowing us to compare different groups. [Colour figure can be viewed at wileyonlinelibrary.com]

maximum (% 1-peak: 86 ± 15% after transfer) and correspondingly the percentage of transients with more complex contractile behaviour dropped (% >1-peak: 14 ± 15% after transfer). This profile was maintained for 24 h, but over the subsequent 2 days, the percentage of grid squares showing single peaks declined back towards the control value (% 1-peak: 53 ± 24% on day 3) and was maintained until day 7 (% 1-peak: 45 ± 10%), while the grid squares displaying more complex behaviour increased correspondingly.

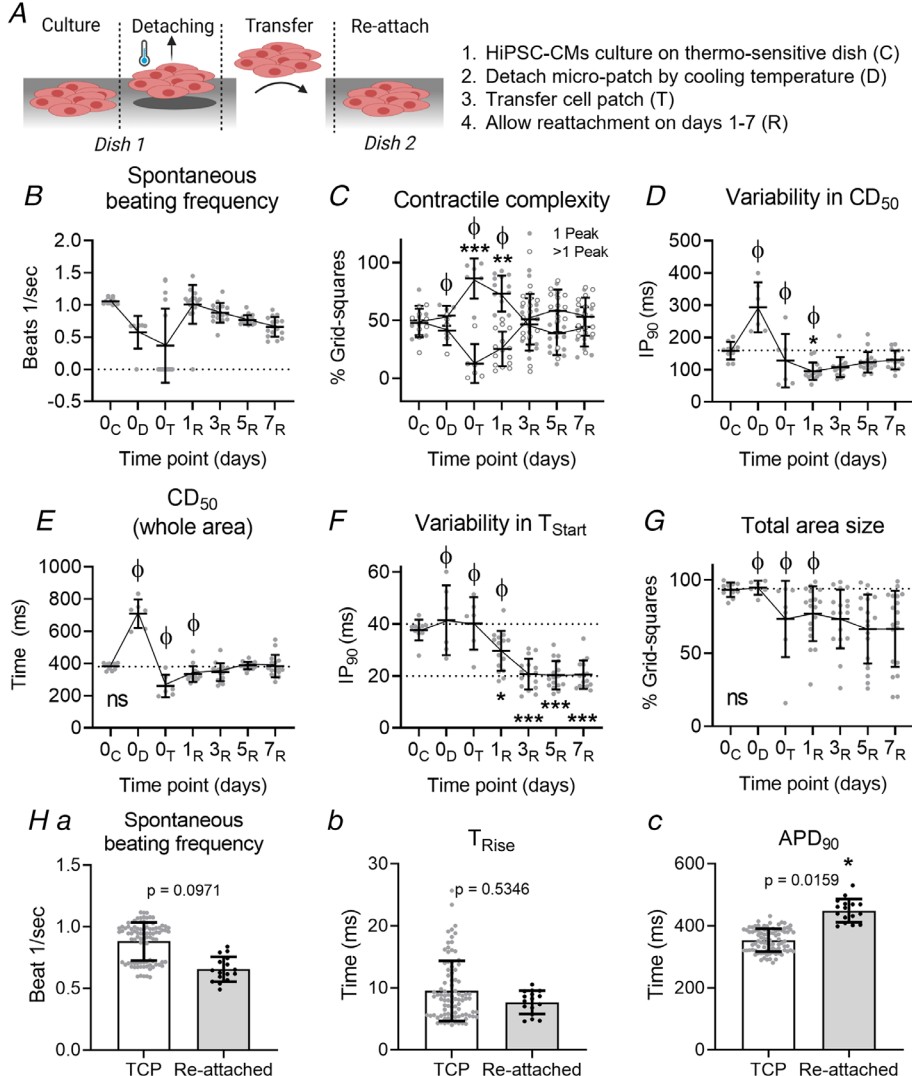

**Figure 8. Contractile behaviour of hiPSC-CMs before detachment, right after detachment and during reattachment**
*A*, animation explaining the experimental procedure of cell culture on a thermosensitive dish (C), cooling the plate down to 25°C, allowing the cell sheet to detach while maintaining cell–cell adhesions (D), transferring the cell sheet to a new culture dish (T) and allowing the cells to reattach to the new culture dish (R). *B*, spontaneous beating frequency of hiPSC-CMs. *C*, the contractile complexity, expressed as the average percentage of grid squares with either single- (black dots) or multiple-peaked (white dots) transients. *D*, the $IP_{90}$ for $CD_{50}$, describing the variation in contractile behaviour. *E*, the $CD_{50}$ taken from the whole area video recording. *F*, the $IP_{90}$ for the $T_{Start}$, describing the level of synchronized contraction within the cell sheet. *G*, the average percentage of grid squares containing beating cells, indicating the size of the cell area. $\phi$ = quiescent patches were excluded from analysis at those time points. Time points were statistically tested against D0 37°C using a nested one-way ANOVA with Dunnett's *post hoc* test. *H*, voltage recordings of reattached (D7) monolayers (Cor.4U (NCardia)) compared with data presented in Fig. 4A as these were from comparable experimental groups. Recordings were made on days 2 (TCP) and 7 after stencil removal. (a) Spontaneous beating frequency. (b) $T_{Rise}$. (c) $APD_{90}$. Groups were compared using a nested, two-sided *t* test. $n_{Experiments}$ = 3, $n_{Samples}$ = 16. A *P* value <0.05 is considered significant. *= *P* < 0.05, **= *P* < 0.01, ***= *P* < 0.001. Absolute values of the *P* values for all comparisons (significant and non-significant) are listed in the online Statistical Supplement. [Colour figure can be viewed at wileyonlinelibrary.com]

Figure 8*D* shows the variability in contractile duration, expressed as $IP_{90}$, before, during and after detachment from the rigid substrate. Before detachment, mean $IP_{90}$ for $CD_{50}$ was 150 ms and this approximately doubled on lowering the temperature, which is most likely caused by the lower temperature reducing the conduction velocity across the 2D culture (Dietrichs *et al.* 2020). Within the first day after detachment, mean $IP_{90}$ for $CD_{50}$ decreased significantly to $95 \pm 27$ ms (*P* value = 0.036) but increased again to $130 \pm 29$ ms on day 7 (Fig. 8*D*). The change in mean $CD_{50}$ across the 2D monolayer is shown in Fig. 8*E*; the change in absolute value parallels the change in variability as shown in Fig. 8*D*, with the $CD_{50}$ values returning to control values by day 7. An indication of the synchronicity of contraction is given by the $IP_{90}$ of $T_{Start}$ values (Fig. 8*F*). There is no change in synchronicity of the activation of the patch on cooling and detaching, but from day 1 to day 7 the $IP_{90}$ values were significantly lower than the control ($IP_{90} = 38 \pm 4$ ms on day 0, $21 \pm 6$ ms on day 3 ($P \leq 0.0001$), $20 \pm 5$ ms on day 7 ($P \leq 0.0001$)). This might be explained by the reduced cell area as shown in Fig. 8*G*. After complete detachment, the monolayer lost its original shape as it dramatically reduced in diameter and became more 3D-like, suggesting improved cell–cell connection and thus the propagation of the electrical signal. Hence, the significant changes in $IP_{90}$ for $T_{Start}$ are most likely caused by the smaller and more 3D-like structure, rather than the rigid substrate.

The electrophysiology of the reattached monolayer was measured on day 7 and showed comparable beating frequency and $T_{Rise}$ to the control values from Fig. 4*A*, which is shown in Figs. 8*H*a and 8*H*b. However, the $APD_{90}$ was significantly increased in reattached monolayers compared with control: $449 \pm 37$ ms *vs.* $354 \pm 37$ ms for reattached and TCP, respectively ($P = 0.016$) (Fig. 8*H*c).

Taken together, these data show that the complex multiphasic contractile behaviour of hiPSC-CMs is almost completely eliminated once the monolayer is detached from the TCP substrate, but reappears during reattachment. Additionally, variability in $CD_{50}$ and the $CD_{50}$ values taken from the entire monolayer, but not the variability in contraction start time ($T_{Start}$), are affected by substrate rigidity as well. These findings suggest that the attachment to the rigid TCP affect the contractile behaviour of hiPSC-CMs.

## The contractile behaviour of hiPSC-CMs seeded on soft substrates

Another approach to investigate the effect of rigid substrates on the contractile behaviour is to seed hiPSC-CMs on a flexible substrate, such as a hydrogel. In this study, fibronectin-coated collagen hydrogels derived from recombinant collagen peptide (FujiFilm Manufacturing Europe B.V, (Tytgat *et al.* 2019)) with a Young's modulus (G') of approximately 20 kPa were compared with fibronectin-coated glass substrates. Figure 9*A* shows the experimental overview; 3 mm diameter stencils were used to prepare a 3 mm circular patch of hiPSC-CMs onto a 6 mm diameter disc of hydrogel (0.3–0.4 mm thick). The stencil was removed after 2 days, and the cells assessed on days 0, 1, 3, 5 and 7.

The spontaneous beating frequency, voltage recordings (day 7) and calcium recordings (day 7) of hiPSC-CMs seeded on either a hydrogel or glass are shown in Fig. 9*B*, Fig. 9*H* and Fig. 9*I*, respectively, and were similar in both groups, indicating that the patches were of comparable viability on both formats.

In Fig. 9*C*, the average percentage of grid squares with single-peaked transients is shown. This data indicates that cells attached to the collagen gel retain a high percentage of single-peaked grid squares, while the cells on glass progressively developed a significant percentage of grid squares with two or more peaks in each contractile event (% 1-peak on day 7 = hydrogel: $90 \pm 4\%$, control: $62 \pm 8\%$, $P < 0.0001$).

No significant differences were observed in the range of contractile duration values, but the average contraction duration ($CD_{50}$) was significantly shorter in cells seeded on the hydrogel on day 0 ($CD_{50}$ on hydrogel: $246 \pm 27$ ms, glass: $291 \pm 25$ ms, $P = 0.006$), day 3 ($CD_{50}$ on hydrogel: $285 \pm 50$ ms, glass: $366 \pm 41$ ms, $P = 0.002$), day 5 ($CD_{50}$ on hydrogel: $285 \pm 42$ ms, glass: $366 \pm 41$ ms, $P < 0.0001$) and day 7 ($CD_{50}$ on hydrogel: $230 \pm 9$ ms, glass: $298 \pm 17$ ms, $P < 0.0001$) (Figs. 9*D* and 9*E*, respectively). No significant differences were observed in cell synchronicity (Fig. 9*F*) or the number of grid squares with contractile events (Fig. 9*G*).

Separate measurements of intracellular $Ca^{2+}$ indicate similar calcium transients between rigid and flexible substrates as shown in Fig. 9*I*. No significant difference between glass and hydrogel was observed for any of the parameters. The only change noted was an approximately 10% increase in $CaT_{50}$ and $CaT_{90}$ of the calcium transient on flexible substrate (Figs. 9*I*b and 9*I*c).

These data replicate the findings seen before and shown during the detachment/reattachment studies, which observed multiphasic time-courses with relatively variable and long $CD_{50}$ values when attached to TCP. In contrast, these data also indicate that a softer substrate almost eliminates the multiphasic time-courses and reduces the $CD_{50}$ values significantly. Interestingly, the variability in $CD_{50}$ and $T_{Start}$ were not significantly different between both groups, indicating that these variables might not be influenced by substrate stiffness, but rather by geometry or cell–ECM adhesions. However, this is speculation. Taken together these data show that the contractile events on a flexible substrate do not exhibit

the complexity that develops when cells are attached to a fixed substrate such as glass or plastic.

### The contractile time-course of single hiPSC-CMs

The origin of the complex behaviour might be differential attachment of the monolayer to the underlying substrate and not the behaviour of individual cells. Evidence for this is provided by the MM records from isolated single hiPSC-CMs on both glass and flexible substrates. As shown in Figs. 10*A* and 10*B*, individual isolated cells on glass have single-peaked contractions as expected from the standard EC-coupling model. This is also observed in isolated cells seeded on the flexible substrate (Figs. 10*C* and 10*D*). Thus, complex contractile behaviour only arises in a mechanically linked sheet of cells on glass or plastic substrates.

Single cells on were beating faster on average compared with the glass substrate (beating frequency: glass $0.69 \pm 0.14$ beats/s, hydrogel $0.89 \pm 0.32$ beats/s, $P = 0.038$)

as shown in Fig. 10*E*. Nonetheless, both the contraction time and relaxation time were not significantly different between both groups (Figs. 10*F* and 10*G*). Further, single cells seeded on a hydrogel had a significantly shorter $CD_{50}$ compared with glass ($CD_{50}$: hydrogel $271 \pm 150$ ms; glass $351 \pm 98$ ms, $P = 0.038$) (Fig. 10*H*), which was also observed in hiPSC-CM monolayers seeded on the hydrogel (Fig. 9*E*). However, it has to be noted that a shorter $CD_{50}$ could also be caused by the faster spontaneous beating rate. Thus, the substrate stiffness affects the contraction duration in both hiPSC-CMs monolayers and single isolated cells.

### Effect of isoprenaline on the contractile complexity of hiPSC-CMs

To investigate whether a flexible substrate alters the actions of inotropic drugs, we incubated cells with the $\beta$-agonist isoprenaline (ISO) (300 nM) and the contractile

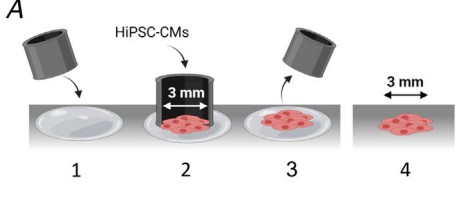

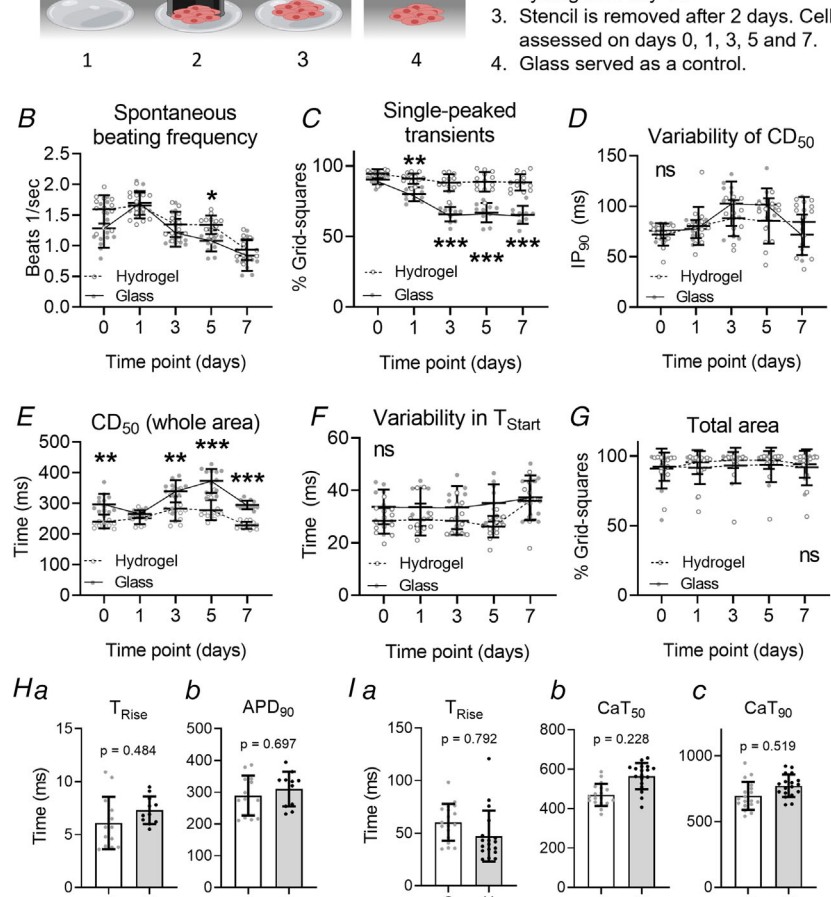

**Figure 9. Investigation of the contractile behaviour of micro-patches seeded on recombinant collagen-like peptide (RCP) hydrogels**

*A*, animation describing the methodology of the hiPSC-CM cell-seeding procedure on the hydrogel. A glass substrate served as a control. Created with BioRender.com (*B*) the average beating frequency of hiPSC-CMs. *C*, the average percentage of grid squares with a single-peaked transient. *D*, the $IP_{90}$ of $CD_{50}$ describing the variation in contractile behaviour. *E*, the $CD_{50}$ taken from the whole area video recording. *F*, the $IP_{90}$ of $T_{Start}$, describing the synchronicity of the contractions. *G*, the percentage of grid squares per recording containing beating cells and thus analysable, indicating the size of the cell area. ICell² (CDI) hiPSC-CMs were used for these experiments. *H*, data obtained from voltage recordings made on day 7 after stencil removal. *I*, data obtained from calcium recordings on day 7 after stencil removal. G = glass. H = hydrogel. $n_{Experiments} = 3$, with 2–6 samples per experiment per time point for each experimental group. Groups were compared with the control value at the same time point using a two-way ANOVA with Sidak's *post hoc* test (*B–G*) or a nested, two-sided *t* test (*H* and *I*), to test statistical significance. A *P* value < 0.05 was considered significant and is indicated with *. Absolute values of the *P* values for all comparisons (significant and non-significant) are listed in the online Statistical Supplement. [Colour figure can be viewed at wileyonlinelibrary.com]

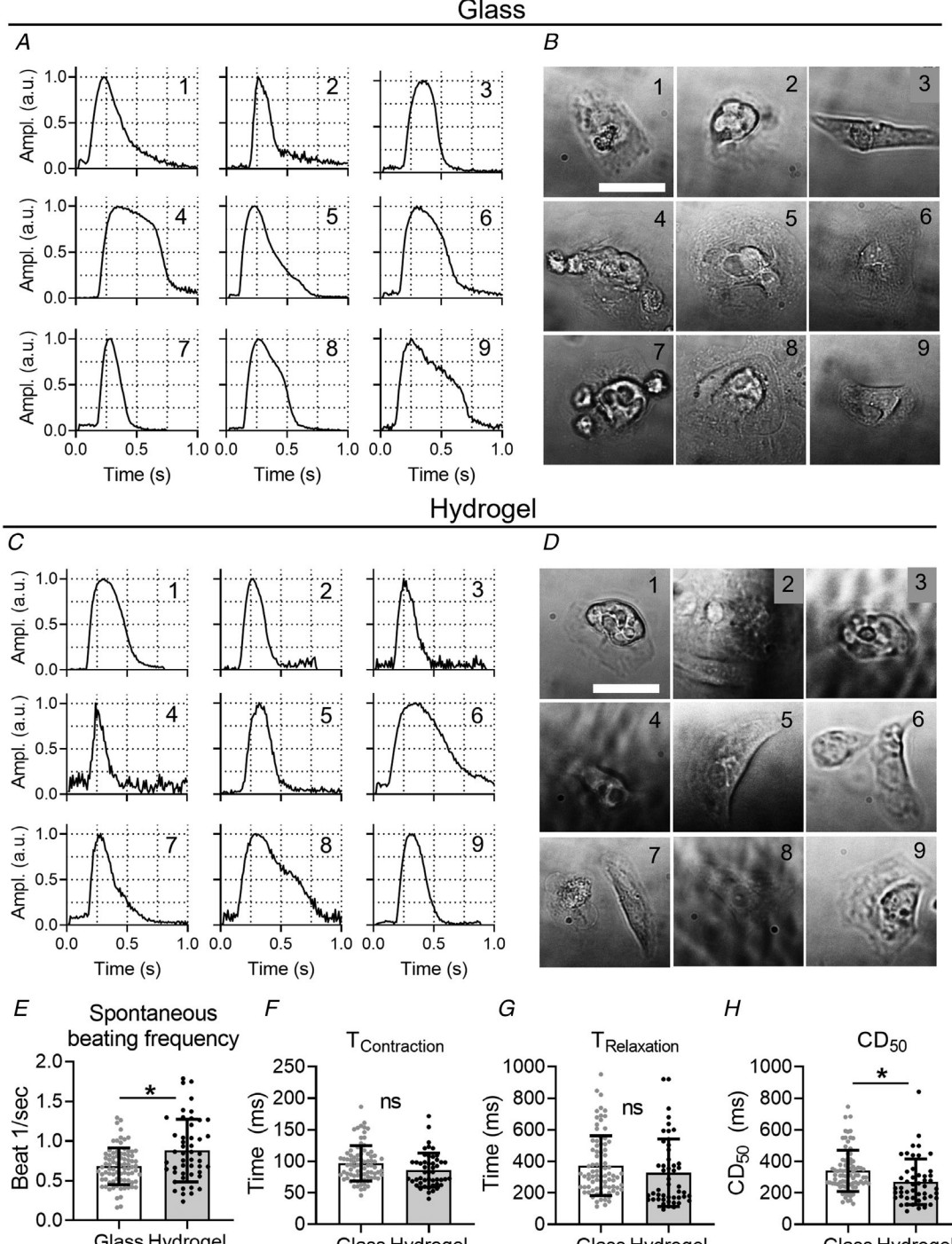

**Figure 10. Single cells only show single-peaked contraction profiles**
Single cell cultures of iPSC-CMs (ICell[2] (CDI)) seeded on either glass (*A* and *B*) or hydrogel (*C* and *D*) substrates. The nine contractility traces with normalized amplitude (*A* and *C*) correspond to the bright-field images of single cells (*B* and *D*). *E*, the spontaneous beating frequency; *F*, the contraction time; *G*, the relaxation time; and *H*, the $CD_{50}$ of cells seeded on either glass or hydrogel. Recordings taken on day 4 or 5 after stencil removal. Scale bar indicates 50 $\mu$m. Statistical analysis was done using a nested unpaired *t* test. A *P* value < 0.05 is indicated with *. $N_{glass} = 82$; $N_{hydrogel} = 48$; $N_{experiments} = 3$. Absolute values of the *P* values for all comparisons (significant and non-significant) are listed in the online Statistical Supplement.

responses were compared with responses on a rigid substrate (glass).

As shown in Fig. 11*A*, the addition of isoprenaline caused a marked increase in spontaneous beating frequency in both hydrogel and glass substrates. In the steady state (5 min), the change in spontaneous frequency of contractions on hydrogel was significantly higher than that observed on glass ($\Delta$ Freq = 0.6 ± 0.1 Hz vs. 0.8 ± 0.1 Hz, $P < 0.0001$). For spatial analysis, the contractile parameters were measured while stimulating the cells at a fixed rate of 1 Hz (10% higher than the maximum spontaneous rate observed in isoprenaline) to avoid the complicating influence of variable spontaneous rates on other contraction parameters.

Figure 11*B* shows the recordings of the 900 (30 × 30) contractile events of a typical patch seeded on glass before and after the addition of ISO. Traces were grouped into single- or multiple-peaked events from which it is evident that a significant number of grid squares contained complex contractile events, the fraction of grid squares

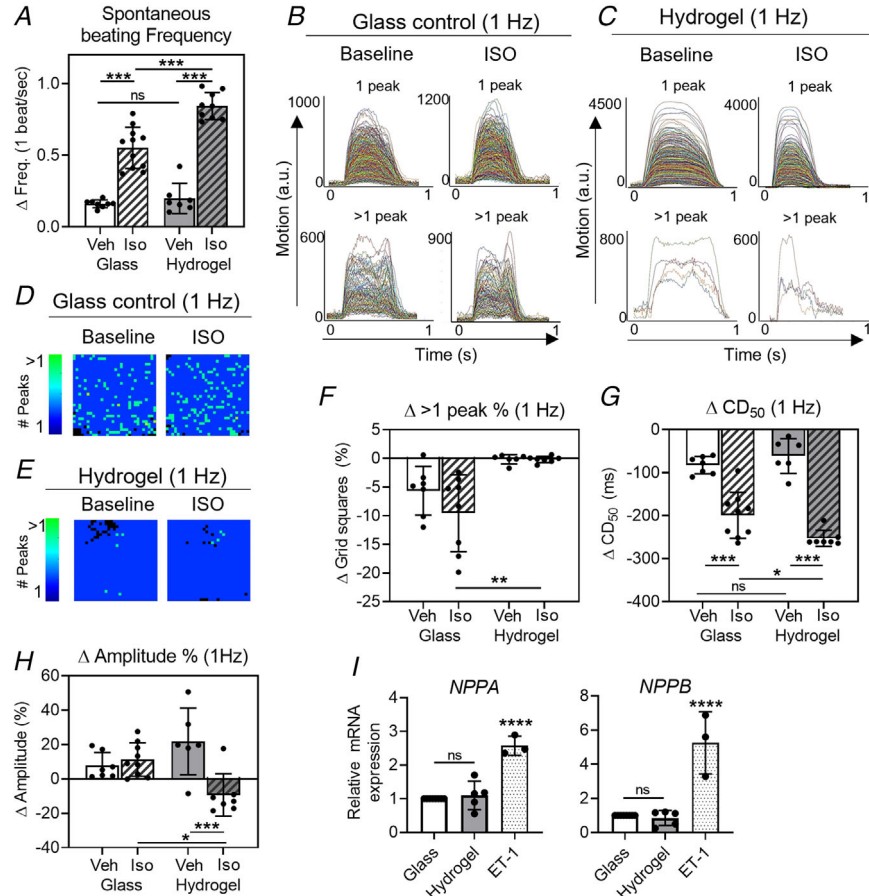

**Figure 11. The effect of isoprenaline on the time-course complexity of hiPSC-CM seeded on different substrate stiffnesses**

HiPSC-CMs (ICell[2] (CDI)) were seeded on either a glass substrate or a hydrogel and incubated with either isoprenaline (ISO, 300 nM) or vehicle (culture medium) for 5 min, where after contractility recordings were made. *A*, the spontaneous beating frequency of hiPSC-CMs. *B* and *C*, contractility traces (900) from the spatial analysis of one representative glass control sample (*B*) and hydrogel (*C*) at baseline and after ISO incubation. Classification of the traces is based on the number of peaks (1 *vs*. >1). *D* and *E*, the distribution of grid squares containing 1-peaked (dark blue) or multiple-peaked transients (light green) at baseline and after ISO incubation for the glass control (*D*) and hydrogel substrate (*E*). *F*, the change in the percentage of grid squares containing multiple-peaked transients compared with the baseline. *G*, the change in average $CD_{50}$ values as compared with the baseline. *H*, the change in amplitude as percentage of the baseline values. All recordings were made on day 5 after stencil removal. *I*, relative gene expression for stress-activated genes *NPPA* and *NPPB*. Endothelin-1 (ET-1) treatment served as a positive control. $n_{Experiments}$ = 3, $n_{Samples}$ = 5 to 8 per group. Groups were compared with any other group using a nested one-way ANOVA with Sidak's *post hoc* test (*A*, *F*, *G* and *H*) or a one-way ANOVA with Tukey's *post hoc* test (*I*). A *P* value < 0.05 was considered significant and is indicated with *. * = $P < 0.05$, ** = $P < 0.01$, *** = $P < 0.001$, **** = $P < 0.0001$. Absolute values of the *P* values for all comparisons (significant and non-significant) are listed in the online Statistical Supplement. [Colour figure can be viewed at wileyonlinelibrary.com]

with complex contractions was relatively unchanged by the addition of ISO. For the equivalent traces from the 2D culture on a hydrogel, which is shown in Fig. 11*C*, notably far fewer areas had complex contraction traces. The spatial distribution of the complex contractile events before and after ISO are shown in Figs. 11*D* and 11*E* and indicate that the position and number of grid squares with >1 peak was approximately the same before and after ISO for both substrates. Figure 11*F* summarizes the change in the percentage of grid squares with complex contractile events. No significant difference was observed on the addition of ISO and hydrogel cultures showed essentially no change in the negligible number of grid squares that demonstrate complex contractile events.

Figure 11*G* indicates that the contraction duration at a fixed stimulation rate (1 Hz) was decreased on addition of ISO in both glass and hydrogel substrates. The extent of the decrease in contraction duration is significantly larger and more consistent on the hydrogel surface ($\Delta CD_{50}$ = hydrogel + ISO: -253 ± 19 ms, glass + ISO: -199 ± 53 ms, $P = 0.037$). The change in the amplitude of the contractile event was also quantified in this set of measurements. As shown in Fig. 11*H*, addition of ISO resulted in no significant changes in the amplitude of the contractile events for either glass or hydrogel.

In summary, the complexity of the contractility of 2D cultures is unaffected by $\beta$-adrenergic stimulation on both substrate types. ISO causes no net inotropic change in 2D cultures of hiPSC-CMs but dramatically shortened the time-course of contraction; an effect that was significantly greater in monolayers of iPSC-CMs on the flexible substrate.

## Relationship between substrate stiffness and stress-activated pathways

External mechanical stresses could enhance stress-activated pathways, including upregulation of genes encoding the well-known hypertrophy markers *NPPA* and *NPPB* (Välimäki *et al.* 2017; Eschenhagen & Carrier, 2019; Saucerman *et al.* 2019; Goetze *et al.* 2020; Pohjolainen *et al.* 2020). Gene expression profiling of hiPSC-CMs seeded on either hydrogel or glass for *NPPA* and *NPPB* showed no statistically significant differences between each group, but both were significantly lower when compared with cells subjected to a hypertrophic stimulus provided by endothelin-1 (ET-1) (fold change *vs.* glass: *NPPA*: hydrogel 1.10 ± 0.42, $P = 0.796$, ET-1 2.57 ± 0.28, $P < 0.0001$; *NPPB*: hydrogel 0.85 ± 0.44, $P = 0.932$, ET-1 5.25 ± 1.82, $P < 0.0001$) (Fig. 11*I*). This indicates that stiff culture substrates do not significantly activate these mechanical stress responses within hiPSC-CMs through the *NPPA* and *NPPB* pathway.

## Computational model to investigate the biological findings

To investigate the observations that the rigid surface is the source of the complex contractile behaviour, a computational model was developed based on previously published descriptions of sarcomere-based contraction models of cardiomyocytes (Rice *et al.* 2008; Timmermann *et al.* 2019).

The computational model of a 1D arrangement of cells was first investigated over a range of conditions; in particular, a different number of cells in the linear array ($N = 3$, $N = 5$ and $N = 10$) and different degrees of cell-to-cell variation in $[Ca^{2+}]_i$, as illustrated in Fig. 12. More than five units result in multiphasic motion that is incompatible with our biological results (Fig. 12). Our hypothesis is that the heterogeneity of the contractile activity in each contractile unit (cell) is the origin of the complex contractile behaviour on stiff matrices. There may be multiple reasons for the variation, but one identified heterogeneity is the cell-to-cell variation in the amplitude of the intracellular calcium transient (Cerignoli *et al.* 2012). Various degrees of heterogeneity of peak $[Ca^{2+}]_i$ were investigated and a moderate level that corresponded with the experimental observations (50–115% of 1.5 $\mu$M) was used (Fig. 12, red boxes).

Figure 13*A* shows the diagram of the model for the arrangement of the cells on the stiff (Fig. 13*A*a) and soft (Fig. 13*A*b) substrates. Here, five contractile units, or cells, are attached to a substrate through a spring constant, *k*, that represents the difference in stiffness for soft ($k = 0.01$) and stiff ($k = 1$) substrates. The same distribution of $[Ca^{2+}]_i$ (medium, 50–115% of 1.5 $\mu$M) was applied to both models to keep all factors identical except for the spring constant (Figs. 13*B*a and 13*B*b). Here, the cells on the soft substrate ($k = 0.01$) practically all shortened during the increase in $[Ca^{2+}]_i$ (Fig. 13*B*d). However, the cells on the rigid substrate ($k = 1$) showed a distinct pattern of both shortening and elongation (Fig. 13*B*c). As shown in Fig. 13*B*e, this resulted in a similar complex motion that we observed in our hiPSC-CM cultures seeded on fixed substrates, including multiple-phasic contraction profiles and delayed transients. In contrast, the transients resulting from the model for the soft substrate only showed twitch-like motion without delay (Fig. 13*B*f). Simulating the same model with other randomized distributions of $[Ca^{2+}]_i$, had similar outcomes, which can be seen in Figs. 13*C*a and 13*C*b for stiff and soft substrates, respectively. The data obtained using this model support the hypothesis that the complex motion signal arises from cell-to-cell interactions within small groups of cells that are not individually attached to a stiff matrix. Flexible matrices with a stiffness comparable to a cell

allows shortening of the sarcomere within the groups of cells and prevents significant cell-to-cell variation in contraction.

## Discussion and conclusions

In recent years, it has become clearer that the mechanical properties of the ECM and the functionality of CMs are strongly related and that the flexibility of the myocardial ECM is key for healthy cardiac physiology (Li *et al.* 2014; Saucerman *et al.* 2019; Ward & Iskratsch, 2020). Yet cardiac cells, including hiPSC-CMs, are traditionally seeded on rigid surfaces, like glass or TCP (Blinova *et al.* 2018; van Meer *et al.* 2019). In this study, we showed that spatial analysis of hiPSC-CM movement exhibited a complex multiphasic contractile transient when cultured on rigid substrates and that this complex behaviour was absent in detached monolayers or when cells were cultured on a flexible substrate and a more physiological 'twitch-like' contraction dominates. As a consequence of the flexible substrate, the lusitropic and chronotropic effects of ISO were enhanced and more consistent than those seen on the rigid substrate.

**Figure 12. The effect of variation in intracellular calcium and number of contractile units on the computed motion by the model presented**
*A*, average motion from 15 sets of simulations with *N* = 3, 5 and 10 contractile units, with *k* = 0.01 and randomly assigned maximum calcium concentrations within the ranges: (*a*) a minimal range of 95–100%, (*b*) a low level of variation, 75–105%, (*c*) a medium level of variation, 50–115% and (*d*) the highest level of variation, 25–125%. The red box indicates the set seen in the main body of the text; specifically, the case with *N* = 5 and a medium level of variation of the maximum calcium. The average of the 15 sets of averages is plotted with the thick black line and are plotted in *B*. *C*, average motion from 15 sets of simulations with *N* = 3, 5 and 10 contractile units, with *k* = 1 and randomly assigned maximum calcium concentrations within the ranges: (*a*) a minimal range of 95–100%, (*b*) a low level of variation, 75–105%, (*c*) a medium level of variation, 50–115% and (*d*) the highest level of variation, 25–125%. The red box indicates the set seen in the main body of the text; specifically, the case with *N* = 5 and a medium level of variation of the maximum calcium. The average of the 15 sets of averages is plotted with the thick black line and is plotted in *D*. [Colour figure can be viewed at wileyonlinelibrary.com]

## The origin of complex contraction time-courses on rigid substrates

The multiple-peaked contraction profiles analysed using the MM algorithm represented marked phasic changes in the movement of areas of the monolayer with respect to the fixed surface during the contraction (Figs. 5 and 7 and Supplementary videos S1 and S2). This indicates marked heterogeneity of cell-to-cell motion across areas of the monolayer in the contractile phase. This heterogeneity occurs despite homogeneous AP signals across the same area of monolayer. The uniformity of APs indicates uniform $Ca^{2+}$ fluxes; in particular, influx via the L-type $Ca^{2+}$ channel and efflux, predominantly through the $Na^+/Ca^{2+}$ exchanger (NCX). As verified by separate intracellular $Ca^{2+}$ measurements (Fig. 9I) there was no marked difference in $Ca^{2+}$ signalling between the two substrates. Radically different $Ca^{2+}$ buffering may have occurred in regions experiencing different movement characteristics, which may alter the local free $Ca^{2+}$ transients (Cerignoli *et al.* 2012). However, this is not thought to be significant under these conditions as there was no evidence for significant differences in the $Ca^{2+}$ transients or AP recordings. It may be that the microscopic gradients of intracellular $Ca^{2+}$ vary over tens of milliseconds in regions of the 2D culture experiencing dramatic motion changes. Unfortunately, rapid spatial analysis of $Ca^{2+}$ signals uncontaminated by movement artefacts is technically very difficult to achieve under these conditions. Nevertheless, the overall similarities of the AP and $Ca^{2+}$ transients would indicate that the average cellular $Ca^{2+}$ transients on rigid and flexible substrates did not differ significantly.

In the contractile phase induced by the near-synchronous APs, the complex movement events appear to arise from differential adhesion between the substrate and the monolayer. In elegant studies of intracellular forces and movement in cell pairs, Maruthamuthu *et al.* (2011) showed that uniform forces were experienced between cell–cell and cell–substrate interfaces, resulting in differential movements across these two types of interface depending on their relative stiffness (Maruthamuthu

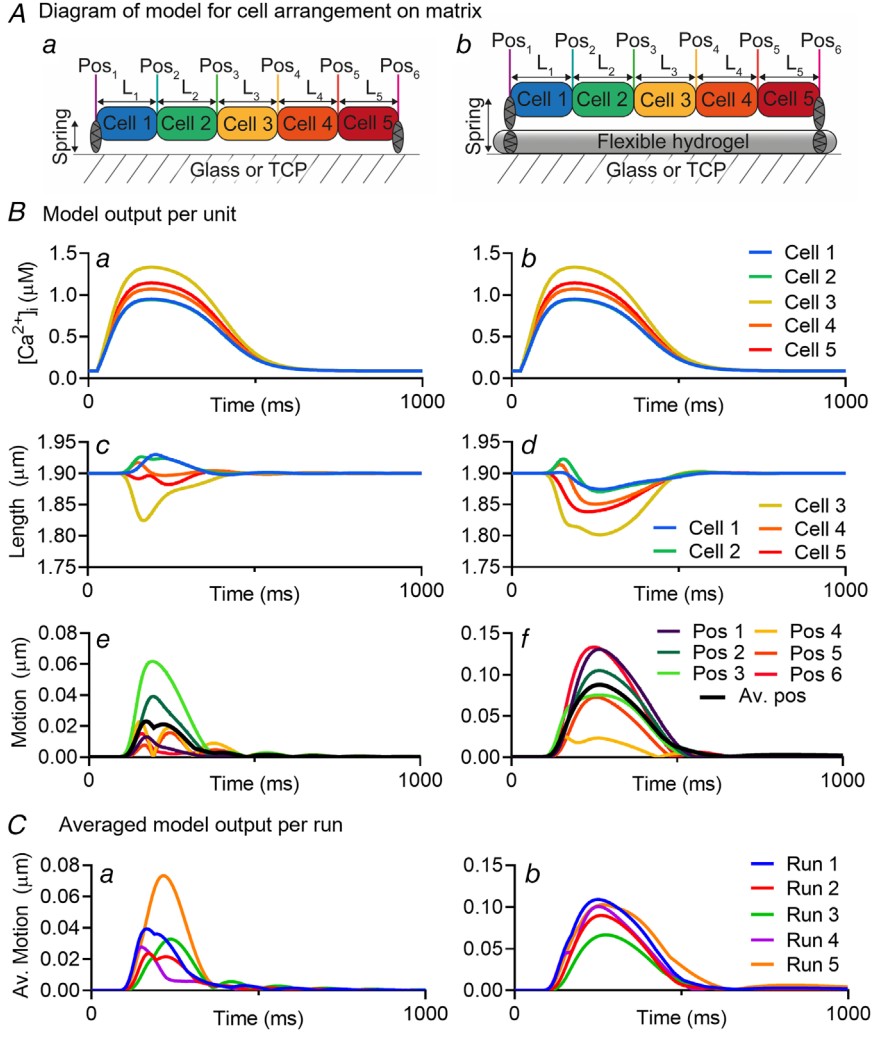

**Figure 13. Computational model showing the complex contractile behaviour as seen in biological samples**

*A*, diagrams showing the model for the cell arrangement on the stiff (*a*) and soft (*b*) substrates. The stiff substrate is represented by a spring at each end with value of the spring constant equal to $k = 1$ and soft substrate is represented by a pair of springs at each end with value of the effective spring constant equal to $k = 0.01$. *B*, graphs showing the output per unit (cell) for intracellular calcium ($\mu$M) (*a* and *b*), cell length ($\mu$m) (*c* and *d*) and the displacement of the ends of a cell from their initial positions (motion) ($\mu$m) (*e* and *f*) of hiPSC-CMs seeded on stiff (*a*, *c*, *e*) and soft (*b*, *d*, *f*) substrates. Colour schemes of panel (*B*) correspond to the diagram in panel (*A*). The black line in panels *Be* and *Bf* represents the average position (Av. Pos) of all the positions (1–6) combined. *C*, the averaged model output per run for five different runs of the same model, but different randomized calcium distribution for each cell. Note that run 2 in both panels *Ca* and *Cb* correspond to the average position of panels *Be* and *Bf*, respectively. Pos denotes position. L denotes cell length or unit length. [Colour figure can be viewed at wileyonlinelibrary.com]

*et al.* 2011). If this model of cell mechanics equally applied to the CM monolayer it would predict dramatically different cell-to-cell movements depending on the underlying adhesion due to the very different local stiffness characteristics. In contrast, the integration of cell-to-matrix forces produced by an underlying flexible substrate would in theory prevent dramatic spatial gradients and potentially explain the absence of heterogeneous movement on the collagen substrate reported in this study.

### Contraction time-course changes on flexible substrates

This study established that hiPSC-CMs could be cultured stably on the surface of the RCP hydrogel for several days. The contractility and electrophysiolgel measurements showed no detrimental effects of these culture conditions for over a week. Our study showed that the level of synchronicity across the monolayer was similar on both stiff and flexible substrates (Fig. 9*F*). However, the contractile duration was shorter in hiPSC-CM monolayers seeded on the flexible substrate (Fig. 9*E*) and was close to that measured from single isolated hiPSC-CMs (Fig. 10*H*). This is in line with other studies that showed that contraction and relaxation velocity were impaired by matrix stiffening, including on glass, in isolated adult rat CMs (van Deel *et al.* 2017). Interestingly, this was reversed when cells were transferred from stiff (100 kPa) to soft (15 kPa) substrates, indicating that the cellular adaptations to mechanical changes are not permanent. Additionally, Sewanan *et al.* (2019) found that hiPSC-CMs seeded on 3D decellularized ECM from stiffer hypertrophic myocardium resulted in longer relaxation time compared with ECM from healthy hearts (Sewanan *et al.* 2019). One potential cause for the difference in mechanical behaviour between the two substrates may be the degree of adhesion. The current study was unable to measure the degree of adhesion across the two substrate types, but as the model showed, the difference in adhesion was not required to explain the difference in contractile behaviour. Therefore, it is unlikely that the level of cellular adhesion is a significant factor in the differential contractile responses.

Several studies have examined the contractile events of single cells cultured on flexible substrates and associated the change over time with changes in sarcomere alignment and other aspects of cell culture that reflected cell maturation (Engler *et al.* 2008; Jacot *et al.* 2008; Ribeiro *et al.* 2015, 2020). None of these studies reported the complex phases of contraction as reported here and, similarly, these phases were not present in recordings of single isolated hiPSC-CMs (Figs. 10*A*–10*D*). It is unlikely that maturation rather than mechanical factors provides an explanation for the more physiological time-course of

monolayers seen on the flexible substrate in this study, because these changes can occur acutely (within 1 day) by simply detaching the monolayer from the underlying substrate (Fig. 8). Furthermore, there were no major changes in electrophysiology (APD) and contraction time-course ($CD_{50}$) over time on flexible substrates, supporting the lack of an ongoing maturation process. Interestingly, reattached monolayers did display a relatively long $APD_{90}$ ($\sim$450 ms), compared with those on a rigid substrate, especially considering their relatively fast spontaneous beating frequency of almost 2 Hz (Fig. 8*H*c). The reason for this is unclear and might be related to the more 3D-like structure and associated electrical syncytium that develops after detachment (Lemoine *et al.* 2017; Lemme *et al.* 2018).

In addition to this, gene expression profiles for the main stress-activated hypertrophy markers *NPPA* and *NPPB* indicated that cells on either substrate experienced comparable stress levels (Fig. 11*I*). Therefore, it is unlikely that the multiphasic contractile profiles cause upregulation of stress-activated pathways nor that activation of the *NPPA* and *NPPB* pathways result in heterogeneous contractile profiles within the monolayer. However, *NPPA* and *NPPB* expression could be affected by different culture conditions, such as lower cell-seeding densities, changing the outcome. Moreover, *NPPA* and *NPPB* represent a small section of stress-activated pathways. Screening of other genes involved in stress-activated pathways, such as myosin heavy chain (*MYH*), connexin 43 (*GJA1*) and skeletal $\alpha$-actin (*ACTA1*), could provide more insight. However, *NPPA* and *NPPB* share large parts of their upstream pathways with these factors and that is sufficient for this study (Saucerman *et al.* 2019; Pohjolainen *et al.* 2020).

### *In silico* model

The hypothesis that the unphysiological complex contractile behaviour is caused by the interaction of cells with the rigid matrix was tested with a mathematical model. In 1D arrays of up to five cells, the model shows that differences in peak calcium between cells generate different contractile forces at the level of the sarcomere. This results in sarcomere and cell shortening in cells with high $[Ca^{2+}]_i$, while adjacent cells with lower $[Ca^{2+}]_i$ are stretched, increasing both sarcomere- and cell length. Due to the dome-shaped relationship of the sarcomere length–tension curve, as the cells shorten, these forces cannot be maintained and result in spontaneous stretching as the forces in adjacent cells become greater. These interactions occur during the phase of high calcium within one contraction and generate multiphasic changes in cell length within each cluster. This behaviour contrasts with that of a flexible substrate where the overall array of

cells shortens uniformly because the underlying substrate shortens in parallel (Fig. 13). This difference in behaviour does not critically depend on the extensive differences in cell-to-cell calcium, but more complex behaviour arises when the cell array size consists of more than five cells (Fig. 12). Thus, the *in silico* modelling supports the view that the multiphasic contractile behaviour arises from the cell-to-cell interactions of small clusters of hiPSC-CM differentially attached to the underlying stiff matrix and that the matrix stiffnesses approximating that of the cells are required for more physiological contraction profiles.

### Responses of hiPSC-CMs to ISO on rigid and flexible substrates

In an initial effort to examine the consequences of the flexible substrate on the response of hiPSC-CMs on an inotropic intervention, the features of the contractile signal were examined before and after 300 nM isoprenaline. As shown in Fig. 11, the complexity of the contractile event across the monolayer was not altered by the addition of isoprenaline despite the increase in spontaneous frequency. Stimulation at a fixed rate allowed the comparison of the contractile time-course before and after isoprenaline and demonstrated the absence of an inotropic effect but the presence of a clear lusitropic action of the drug. Also, the addition of ISO decreased the duration of contraction because of the increased rate of relaxation. Similar results of the addition of ISO have been observed in previous studies using hiPSC-CMs and hESC-CMs in media with normal extracellular $Ca^{2+}$ (Pillekamp *et al.* 2012; Chen *et al.* 2015; Lewandowski *et al.* 2018). The underlying mechanism is thought to be the summed effects of $\beta$-adrenoreceptor-mediated phosphorylation of L-type $Ca^{2+}$ channel, contractile protein troponin-I and the SR uptake regulatory protein phospholamban. The combined result of these processes in iPSC-CMs is a limited inotropic action due to reduced contribution of the sarcoplasmic reticulum in the response due to immature status of the SR (Liu *et al.* 2007; Pillekamp *et al.* 2012; Chen *et al.* 2015). The response of the monolayer on flexible substrates was like that on rigid ones with an increase in spontaneous frequency, neutral inotropic change and shorter contractile duration. Interestingly, the relative increase in both the spontaneous frequency and the decreased $CD_{50}$ at a fixed rate were greater on the flexible substrates than compared with the rigid substrate. The mechanism underlying these relatively larger chronotropic and lusitropic actions on a flexible substrate is unclear. One possibility is that the greater chronotropic and lusitropic actions are an outcome of the less complex events during the contractile phase on the flexible substrates, which allows more rapid mechanical events on the monolayer, but altered sensitivity to ISO

or altered intracellular pathways may also contribute. Further work is required to fully understand the consequences of the flexible substrate on the inotropic action of drugs on the more physiological, flexible substrates.

### Overall conclusion and future implications

In summary, *in silico* modelling supports experimental data indicating that the multiphasic contraction profiles arise from the attachment of hiPSC-CM groups on stiff matrices such as glass and TCP. The overall effects of the mechanically flexible substrate on hiPSC-CMs were (1) the simpler and more physiological contractile events than those seen on a fixed substrate, (2) a shorter overall duration of contraction, and (3) enhanced chronotropic and lusitropic effects on the addition of ISO. This work highlights the importance of choosing the appropriate matrices for hiPSC-CM culture. These findings are important for both basic research and cardiotoxicity studies that use hiPSC-CMs as a stable reliable model of human myocardium.

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

# Additional information

## Data availability statement

The raw and processed data required to reproduce these findings can be shared upon reasonable request.

## Competing interests

The authors declare that they have no known competing financial interests or personal relationships that could have appeared to influence the work reported in this paper.

## Funding

This work was supported by the British Heart Foundation (BHF grant number FS/16/55/32731), the UK Engineering and Physical Sciences Research Council (EPSRC grant numbers EP/N014642/1, EP/S030875/1, EP/T017899/1), the Academy of Finland (grants 321564 and 328909), Sigrid Jusélius Foundation and the Finnish Foundation for Cardiovascular Research.

## Author contributions

E.H.: Conceptualization, Investigation, Writing-Original draft, Visualization, Formal analysis. P.M.: Methodology, Software, Data curation, Writing-Original draft. H.G.: Methodology, Writing-Review and Editing. R.S.: Methodology, Writing-Review and Editing. L.P.: Investigation. V.T.: validation, conceptualization. H.R.: conceptualization. F.B.: Software, Conceptualization, Data curation, Writing-Review and editing. N.G.: Writing-Review and editing, Supervision. G.S.: Conceptualization, Writing-Original draft preparation, Super-vision, Project administration. All authors have read and approved the final version of this manuscript and agree to be accountable for all aspects of the work in ensuring that questions related to the accuracy or integrity of any part of the work are appropriately investigated and resolved. All persons designated as authors qualify for authorship, and all those who qualify for authorship are listed.

## Acknowledgements

The authors would like to thank FujiFilm Manufacturing Europe B.V. for providing the RCP-MA for the hydrogels, with a special thanks to Dr Bas Kluijtmans and Dr Suzan van Dongen for their support.

The authors would like to thank Dr Xie He for manufacturing the stencils used in these studies, Dr Ana Da Silva Costa for her advice on hiPSC-CM culture and Aileen Rankin and Annika Korvenpää for their technical assistance.

## Keywords

cardiac physiology, mathematical model, pharmaceutical assay, recombinant collagen polymer, substrate rigidity

# Supporting information

Additional supporting information can be found online in the Supporting Information section at the end of the HTML view of the article. Supporting information files available:

**Peer Review History**
**Statistical Summary Document**
**Supplemental Video 1**
**Supplemental Video 2**

