## [Peer Review History · The Journal of Physiology]

Conventional rigid 2D substrates cause multiphasic contractile signals in monolayers of human induced pluripotent stem cell derived cardiomyocytes

Eline Huethorst, Peter Mortensen, Radostin D Simitev, Hao Gao, Lotta Pohjolainen, Virpi Talman, Heikki Ruskoaho, Francis L Burton, Nikolaj Gadegaard, and Godfrey L Smith

DOI: 10.1113/JP282228

Corresponding author(s): Godfrey Smith (Godfrey.Smith@glasgow.ac.uk)

The following individual(s) involved in review of this submission have agreed to reveal their identity: Wolfram-Hubertus Zimmermann (Referee #1)

Review Timeline:

Submission Date:	10-Aug-2021
Editorial Decision:	07-Sep-2021
Revision Received:	21-Sep-2021
Editorial Decision:	12-Oct-2021
Revision Received:	27-Oct-2021
Accepted:	03-Nov-2021

Senior Editor: Bjorn Knollmann

Reviewing Editor: Michael Shattock

Transaction Report:

Dear Professor Smith,

Re: JP-RP-2021-282228 "Conventional rigid 2D substrates cause multiphasic contractile signals in monolayers of human induced pluripotent stem cell derived cardiomyocytes" by Eline Huethorst, Peter Mortensen, Radostin D Simitev, Hao Gao, Lotta Pohjolainen, Virpi Talman, Heikki Ruskoaho, Francis L Burton, Nikolaj Gadegaard, and Godfrey L Smith

Thank you for submitting your manuscript to The Journal of Physiology. It has been assessed by a Reviewing Editor and by 2 expert Referees and I am pleased to tell you that it is considered to be acceptable for publication following satisfactory revision.

The reports are copied at the end of this email. Please address all of the points and incorporate all requested revisions, or explain in your Response to Referees why a change has not been made.

NEW POLICY: In order to improve the transparency of its peer review process The Journal of Physiology publishes online as supporting information the peer review history of all articles accepted for publication. Readers will have access to decision letters, including all Editors' comments and referee reports, for each version of the manuscript and any author responses to peer review comments. Referees can decide whether or not they wish to be named on the peer review history document.

Authors are asked to use The Journal's premium BioRender (<https://biorender.com/>) account to create/redrawn their Abstract Figures. Information on how to access The Journal's premium BioRender account is here: <https://physoc.onlinelibrary.wiley.com/journal/14697793/biorender-access> and authors are expected to use this service. This will enable Authors to download high-resolution versions of their figures.

I hope you will find the comments helpful and have no difficulty returning your revisions within 4 weeks.

Your revised manuscript should be submitted online using the links in Author Tasks Link Not Available.

Any image files uploaded with the previous version are retained on the system. Please ensure you replace or remove all files that have been revised.

REVISION CHECKLIST:

- Article file, including any tables and figure legends, must be in an editable format (eg Word)
- Abstract figure file (see above)
- Statistical Summary Document
- Upload each figure as a separate high quality file
- Upload a full Response to Referees, including a response to any Senior and Reviewing Editor Comments;
- Upload a copy of the manuscript with the changes highlighted.

- A potential 'Cover Art' file for consideration as the Issue's cover image;
- Appropriate Supporting Information (Video, audio or data set https://jp.msubmit.net/cgi-bin/main.plex?form_type=display_requirements#supp).

To create your 'Response to Referees' copy all the reports, including any comments from the Senior and Reviewing Editors, into a Word, or similar, file and respond to each point in colour or CAPITALS and upload this when you submit your revision.

I look forward to receiving your revised submission.

If you have any queries please reply to this email and staff will be happy to assist.

Yours sincerely,

Bjorn Knollmann
Senior Editor
The Journal of Physiology

EDITOR COMMENTS

Reviewing Editor:

Comments for Authors to ensure the paper complies with the Statistics Policy:

Please ensure that your data presentation meet the requirements of our Statistics Policy. In particular please include all raw data as data points on graphs as well as means and SDs. As data are also presented as a number of observations from a small number of experiments, please also use appropriate hierarchical statistical analysis to prevent clustering bias.

Comments to the Author:

Both reviewers thought this an interesting and useful study but had some reservations. Please address the comments of reviewer 1.

Senior Editor:

Comments for Authors to ensure the paper complies with the Statistics Policy:

As per reviewing editors comments

Comments to the Author:

Careful study on a topic of current interest in the field. Please address the concerns raised by reviewer one. In particular, the claim that substrate stiffness does not play a role in stress pathway activation needs to be toned down. This was not the goal of the current study, and has been examined by others in much more detail. Given the tenting that occurred in the 2D monolayer on the "stiff" substrate, the substrate stiffness is not defined.

REFEREE COMMENTS

Referee #1:

Huethorst and colleagues report differential mechanical properties of iPSC-derived cardiomyocyte monolayers. If cultured at high cell density on rigid substrates CM monolayers display multiphasic motion. Which is absent in single cells and in high density monolayers on soft substrates. The study comprises experimental results obtained from two commercial iPSC-CM sources and describes a model for linear arrays of cardiomyocytes on stiff and soft substrates. Finally, the authors conclude that basic research and cardiotoxicity studies would benefit from iPSC-CM cultures on appropriate, i.e., flexible, substrates.

Comments:

Motion in monolayers depends on pacemaker activity and spread of excitation. Dense cardiomyocyte monolayers tend to detach from stiff substrates, whereas adhesion on soft substrates tends to be more uniform. Accordingly, synchronicity in monolayers on soft substrates is higher. Please, discuss potential differences in cardiomyocyte adhesion and in particular the contribution to monolayer detachment to the multiphasic motion. Alternatively, provide data in support of similar adhesion on soft and stiff substrates.

Table 1: please add the quantities of truly retained cells after plating; according to Suppl. Fig 1E this is 20-50% of the seeded cells. Please specify whether the company-provided cell number is total, viable or platable cells. The finding of multiphasic contractions on stiff substrates is clearly dependent on cell density and culture conditions. Please also add information on cell purity in the methods section.

The claim that substrate stiffness does not play a role in stress pathway activation requires a re-consideration. It is quite likely that stress pathway activation will differ depending on cell density.

Minor points:

Page 7: "However, these studies were mostly done with single cells, and overlooked the additional role of cell-cell coupling and intercellular force transmission (Yonemura et al., 2010; Monemian Esfahani et al., 2019)." - I would not state that this was "overlooked". Single cell studies are done to avoid the ensemble phenomena described in the dense monolayers on stiff substrates by Huethorst et al.; in addition, hypertrophy/stress pathway induction studies are typically performed in responsive low density cultures

Page 8, line 10: "... or hydrogel substrates (10 µg/ml, bovine, Gibco) in ..." - please specify that you were using bovine collagen

Referee #2:

Since there is interest in the effects of drugs on stem cell derived cardiomyocyte contractility, it is important to know what conditions will give the best contractile behavior of the cardiomyocytes. In these studies, they show that the desired twitch-like transients are present in monolayers on soft substrates. The imaging patterns of the undesirable multiphasic contractions are quite distinct from those of the twitch-like behaviors and the documentation of the differences on soft vs. rigid substrates is quite clear. Thus, this is a well-documented paper that identifies which substrates can be used for drug studies of stem-cell derived cardiomyocytes. If this is an important finding for the field, I can recommend publication.

END OF COMMENTS

Confidential Review

10-Aug-2021

Response to reviewers:

We would like to thank all reviewers for their time and their comments in order to improve the manuscript. We have amended the manuscript accordingly and all responses are written below in red underneath the corresponding comment.

In addition to this, we have amended the figures to show all raw data points along with the means and SDs to comply with the Statistics Policy of the Journal of Physiology. Furthermore, we have re-analysed the data using a nested method where applicable. This is all written down in the Statistical Summary Table that is included with the submission.

Referee #1:

Comments:

1. *Motion in monolayers depends on pacemaker activity and spread of excitation. Dense cardiomyocyte monolayers tend to detach from stiff substrates, whereas adhesion on soft substrates tends to be more uniform. Accordingly, synchronicity in monolayers on soft substrates is higher. Please, discuss potential differences in cardiomyocyte adhesion and in particular the contribution to monolayer detachment to the multiphasic motion. Alternatively, provide data in support of similar adhesion on soft and stiff substrates.*

The reviewer makes a good point and one we did consider. The measurements would need to assay the density of Cell Adhesion Complexes, and also assess the extent to which CACs were attached to the substrate. This is currently beyond our technology, but we have complied with the reviewer's request and added discussion on the role of cell adhesion and its contribution to multiphasic motion as described. (page 27, lines 14-15 and lines 24-29)

2. *Table 1: please add the quantities of truly retained cells after plating; according to Suppl. Fig 1E this is 20-50% of the seeded cells. Please specify whether the company-provided cell number is total, viable or platable cells. The finding of multiphasic contractions on stiff substrates is clearly dependent on cell density and culture conditions. Please also add information on cell purity in the methods section.*

We have complied with the reviewer's request and supplied the quantities of retained cells after plating in the modified version of Table1 (page 38). Furthermore, we clarified that we are stating the plated cells and the viable cells, based on the percentages of Supplementary Figure 1. We also amended the legend accordingly. Then, we stated the purity of the two cell lines on page 5, lines 23-25.

3. *The claim that substrate stiffness does not play a role in stress pathway activation requires a re-consideration. It is quite likely that stress pathway activation will differ depending on cell density.*

We agree with the reviewer and have edited the Results section and Discussion to reflect this point. Mainly, through making our conclusion softer and emphasising that the mechanical stress responses

within hiPSC-CMs through the NPPA and NPPB pathways are not upregulated. (page 21, line 14-16, and page 28, lines 5-16)

Minor points:

4. Page 7: "However, these studies were mostly done with single cells, and overlooked the additional role of cell-cell coupling and intercellular force transmission (Yonemura et al., 2010; Monemian Esfahani et al., 2019)." - I would not state that this was "overlooked". Single cell studies are done to avoid the ensemble phenomena described in the dense monolayers on stiff substrates by Huethorst et al.; in addition, hypertrophy/stress pathway induction studies are typically performed in responsive low density cultures

The reviewer makes a good point, the word "overlooked" was not appropriate. In line with the reviewer's comments, we have reworded that section of the Introduction to: "However, these studies were mostly done with single cells, and therefore omitted the additional role of cell-cell coupling and intercellular force transmission, which could be important for contractile dynamics across the monolayer". (page 4, line 25-28)

5. Page 8, line 10: "... or hydrogel substrates (10 µg/ml, bovine, Gibco) in ..." - please specify that you were using bovine collagen

We realize that this was not clearly stated. We used bovine fibronectin and human collagen. This has been clarified on page 5, lines 25-26 and on page 8, line 11.

Referee #2:

Since there is interest in the effects of drugs on stem cell derived cardiomyocyte contractility, it is important to know what conditions will give the best contractile behavior of the cardiomyocytes. In these studies, they show that the desired twitch-like transients are present in monolayers on soft substrates. The imaging patterns of the undesirable multiphasic contractions are quite distinct from those of the twitch-like behaviours and the documentation of the differences on soft vs. rigid substrates is quite clear. Thus, this is a well-documented paper that identifies which substrates can be used for drug studies of stem-cell derived cardiomyocytes. If this is an important finding for the field, I can recommend publication.

Thank you for your report. We are pleased you find the study clear and of importance.

Dear Dr Smith,

Re: JP-RP-2021-282228R1 "Conventional rigid 2D substrates cause multiphasic contractile signals in monolayers of human induced pluripotent stem cell derived cardiomyocytes" by Eline Huethorst, Peter Mortensen, Radostin D Simitev, Hao Gao, Lotta Pohjolainen, Virpi Talman, Heikki Ruskoaho, Francis L Burton, Nikolaj Gadegaard, and Godfrey L Smith

Thank you for submitting your manuscript to The Journal of Physiology. It has been assessed by a Reviewing Editor and by 2 expert referees and I am pleased to tell you that it is considered to be acceptable for publication following satisfactory revision.

The reports are copied at the end of this email. Please address all of the points and incorporate all requested revisions, or explain in your Response to Referees why a change has not been made.

NEW POLICY: In order to improve the transparency of its peer review process The Journal of Physiology publishes online as supporting information the peer review history of all articles accepted for publication. Readers will have access to decision letters, including all Editors' comments and referee reports, for each version of the manuscript and any author responses to peer review comments. Referees can decide whether or not they wish to be named on the peer review history document.

I hope you will find the comments helpful and have no difficulty returning revisions within 4 weeks.

If you need to check to make sure that your Methods section conforms to the principles of UK regulations, you may wish to refer to Grundy (2015):
Grundy (2015) J. Physiol. 2015 Jun 15;593(12):2547-9 <https://doi.org/10.1113/JP270818>

Your revised manuscript should be submitted online using the links in Author Tasks Link Not Available. This link is to the Corresponding Author's own account, if this will cause any problems when submitting the revised version please contact us.

The image files from the previous version are retained on the system. Please ensure you replace or remove any files that have been revised.

REVISION CHECKLIST:

- Summary data must be reported as mean {plus minus} SD or 95% confidence interval
- All table and figure legends with summary data must include the statistical test used in the table/figure and sample size
- Figures with summary data bars must include individual data points, or box whisker plots when $n > 30$.
- Article file, including any tables and figure legends, must be in an editable format (eg Word)
- Upload each figure as a separate high quality file
- Upload a full Response to Referees, including a response to any Senior and Reviewing Editor Comments;
- Upload a copy of the manuscript with the changes highlighted.

- A potential 'Cover Art' file for consideration as the Issue's cover image;
- Appropriate Supporting Information (Video, audio or data set https://jp.msubmit.net/cgi-bin/main.plex?form_type=display_requirements#supp).

To create your 'Response to Referees' copy all the reports, including any comments from the Senior and Reviewing Editors, into a Word, or similar, file and respond to each point in colour or CAPITALS and upload this when you submit your revision.

I look forward to receiving your revised submission.

If you have any queries please reply to this email and the Peer Review Coordinator will be pleased to advise.

If revision is not possible, or if you cannot respond to the requests for change, contact us by return email as soon as possible, giving reasons for the difficulties. Withdrawal of the manuscript may be necessary in these circumstances, and instruction will be given on how to proceed. Please note that a paper must be withdrawn before it can be submitted to another journal. If any issues remain unresolved please contact the Publications Office at jphysiol@physoc.org

If you would like help with English language editing, or other article preparation support, Wiley Editing Services offers expert help with English Language Editing, as well as translation, manuscript formatting, and figure formatting at www.wileyauthors.com/eeo/preparation. You can also check out our resources for Preparing Your Article for general guidance about writing and preparing your manuscript at www.wileyauthors.com/eeo/prepresources.

Yours sincerely,

Bjorn Knollmann
Senior Editor
The Journal of Physiology

REQUIRED ITEMS:

Please could authors add...

- 1) a Key points summary
- 2) a statement on competing/conflicting interests

(please refer to:

https://jp.msubmit.net/cgi-bin/main.plex?form_type=display_requirements#Revised%20submissions)

EDITOR COMMENTS

Reviewing Editor:

No further comments.

Senior Editor:

Both reviewers and the reviewing editor were satisfied with the revision. I concur. However, supplemental figures and tables are not allowed per journal policy. Please revise the MS to incorporate all supplemental material (except videos) into the main MS. I would encourage the authors to try to consolidate some of the supplemental figures, or combine some of them with the main MS figures. I realize that this may not be possible for every figure.

REFEREE COMMENTS

Referee #1:

The authors have addressed my critiques. I have no further recommendations.

Referee #2:

I have no further concerns.

END OF COMMENTS

1st Confidential Review

21-Sep-2021

Response to referees

We would like to thank the Senior Editor, and all referees for their time to consider this manuscript for publication. We have made all changes as proposed by the Senior Editor and we hope that you find these changes satisfactory for our manuscript to be published. Our replies are written in red below each recommendation.

REQUIRED ITEMS:

Please could authors add...

- 1) a Key points summary
- 2) a statement on competing/conflicting interests

(please refer to:

https://jp.msubmit.net/cgi-bin/main.plex?form_type=display_requirements#Revised%20submissions)

We have added the key points summary and a statement on competing/conflicting interest to the manuscript (page 3 and page 38, respectively)

Senior Editor:

Both reviewers and the reviewing editor were satisfied with the revision. I concur. However, supplemental figures and tables are not allowed per journal policy. Please revise the MS to incorporate all supplemental material (except videos) into the main MS. I would encourage the authors to try to consolidate some of the supplemental figures, or combine some of them with the main MS figures. I realize that this may not be possible for every figure.

We have included all supplementary figures and the table into the main body of the manuscript in a logical order. To minimize the number of figures, some panels have been combined with other figures and some data has been excluded. All figure numbers have been changed in the text accordingly. Changes are summarized in the following list.

- S Table 1 > Methods section
- S1 > Figure 3, Results section
- S2 > Separated in Figure 1 (Methods) and Figure 4 (Results)
- S3 > Figure 2 Methods
- S4 > Figure 12
- S5 > Figure 6
- S6 > voltage data added to corresponding figures 8 (Detachment study) and figure 9 (Hydrogel study)

- S7 > Amplitude data, excluded as this does not add to the story. Data cannot be compared between groups due to the nature of the algorithm.
- S8 > calcium data added to figure 9 (hydrogel study)
- Supplementary Methods: relocated to the method section, replacing the shorter previous version.

Other alterations to the manuscript

1. Include more details regarding equipment, materials and solutions
2. Changed roman letters to for small latin letters. indicate subpanels
3. Changed the statistical summary sheet accordingly as figure numbers have changed.
4. All alterations are highlighted in yellow in the manuscript.

Dear Dr Smith,

Re: JP-RP-2021-282228R2 "Conventional rigid 2D substrates cause multiphasic contractile signals in monolayers of human induced pluripotent stem cell derived cardiomyocytes" by Eline Huethorst, Peter Mortensen, Radostin D Simitev, Hao Gao, Lotta Pohjolainen, Virpi Talman, Heikki Ruskoaho, Francis L Burton, Nikolaj Gadegaard, and Godfrey L Smith

I am pleased to tell you that your paper has been accepted for publication in The Journal of Physiology.

NEW POLICY: In order to improve the transparency of its peer review process The Journal of Physiology publishes online as supporting information the peer review history of all articles accepted for publication. Readers will have access to decision letters, including all Editors' comments and referee reports, for each version of the manuscript and any author responses to peer review comments. Referees can decide whether or not they wish to be named on the peer review history document.

Are you on Twitter? Once your paper is online, why not share your achievement with your followers. Please tag The Journal (@jphysiol) in any tweets and we will share your accepted paper with our 23,000+ followers!

The last Word version of the paper submitted will be used by the Production Editors to prepare your proof. When this is ready you will receive an email containing a link to Wiley's Online Proofing System. The proof should be checked and corrected as quickly as possible.

Authors should note that it is too late at this point to offer corrections prior to proofing. The accepted version will be published online, ahead of the copy edited and typeset version being made available. Major corrections at proof stage, such as changes to figures, will be referred to the Reviewing Editor for approval before they can be incorporated. Only minor changes, such as to style and consistency, should be made a proof stage. Changes that need to be made after proof stage will usually require a formal correction notice.

All queries at proof stage should be sent to TJP@wiley.com

Yours sincerely,

Bjorn Knollmann
Senior Editor
The Journal of Physiology

P.S. - You can help your research get the attention it deserves! Check out Wiley's free Promotion Guide for best-practice recommendations for promoting your work at www.wileyauthors.com/eoo/guide. And learn more about Wiley Editing Services which offers professional video, design, and writing services to create shareable video abstracts, infographics, conference posters, lay summaries, and research news stories for your research at www.wileyauthors.com/eoo/promotion.

*** IMPORTANT NOTICE ABOUT OPEN ACCESS ***

Information about Open Access policies can be found here <https://physoc.onlinelibrary.wiley.com/hub/access-policies>

To assist authors whose funding agencies mandate public access to published research findings sooner than 12 months after publication The Journal of Physiology allows authors to pay an open access (OA) fee to have their papers made freely available immediately on publication.

You will receive an email from Wiley with details on how to register or log-in to Wiley Authors Services where you will be able to place an OnlineOpen order.

You can check if your funder or institution has a Wiley Open Access Account here <https://authorservices.wiley.com/author-resources/Journal-Authors/licensing-and-open-access/open-access/author-compliance-tool.html>

Your article will be made Open Access upon publication, or as soon as payment is received.

If you wish to put your paper on an OA website such as PMC or UKPMC or your institutional repository within 12 months of publication you must pay the open access fee, which covers the cost of publication.

OnlineOpen articles are deposited in PubMed Central (PMC) and PMC mirror sites. Authors of OnlineOpen articles are

permitted to post the final, published PDF of their article on a website, institutional repository, or other free public server, immediately on publication.

Note to NIH-funded authors: The Journal of Physiology is published on PMC 12 months after publication, NIH-funded authors DO NOT NEED to pay to publish and DO NOT NEED to post their accepted papers on PMC.

EDITOR COMMENTS

Senior Editor:

Excellent and comprehensive work! I would recommend changing the title to better communicate the extensive scope of the paper, which examines substrate dependence of contractile function, electrophysiology and Ca handling in iPSC monolayers.

END OF COMMENTS

2nd Confidential Review

27-Oct-2021